# Triggers and factors associated with moral distress and moral injury in health and social care workers: A systematic review of qualitative studies

Emily S. Beadle[1]*, Agnieszka Walecka[2], Amy V. Sangam[2], Jessica Moorhouse[2], Matthew Winter[1], Helen Munro Wild[1], Daksha Trivedi[3], Annalisa Casarin[1]

1 Department of Psychology, Sport and Geography, School of Life and Medical Sciences, University of Hertfordshire, Hatfield, United Kingdom, 2 Intensive Care Unit, Royal Free Hospital, London, United Kingdom, 3 Centre for Research in Public Health and Community Care, School of Health and Social Work, The University of Hertfordshire, Hatfield, United Kingdom

* e.beadle@herts.ac.uk

**Data Availability Statement:** Data used in data extraction are included in the supplementary data extraction table. All other information is included in

## Abstract

### Objective

At some point in their career, many healthcare workers will experience psychological distress associated with being unable to take morally or ethically correct action, as it aligns with their own values; a phenomenon known as moral distress. Similarly, there are increasing reports of healthcare workers experiencing long-term mental and psychological pain, alongside internal dissonance, known as moral injury. This review examined the triggers and factors associated with moral distress and injury in Health and Social Care Workers (HSCW) employed across a range of clinical settings with the aim of understanding how to mitigate the effects of moral distress and identify potential preventative interventions.

### Methods

A systematic review was conducted and reported according to recommendations from Cochrane and Preferred Reporting Items for Systematic Reviews and Meta-Analyses guidelines. Searches were conducted and updated regularly until January 2024 on 2 main databases (CENTRAL, PubMed) and three specialist databases (Scopus, CINAHL, PsycArticles), alongside hand searches of study registration databases and other systematic reviews reference lists. Eligible studies included a HSCW sample, explored moral distress/ injury as a main aim, and were written in English or Italian. Verbatim quotes were extracted, and article quality was assessed via the CASP toolkit. Thematic analysis was conducted to identify patterns and arrange codes into themes. Specific factors like culture and diversity were explored, and the effects of exceptional circumstances like the pandemic.

### Results

Fifty-one reports of 49 studies were included in the review. Causes and triggers were categorised under three domains: individual, social, and organisational. At the individual level,

the manuscript or available from the reviewed articles themselves.

**Funding:** This project has been supported by Quality Research funding from the Centre for Health Services and Clinical Research, Department of Psychology, Sport and Geography, University of Hertfordshire. Awarding authors (EB, AC). Funders played no role in the study design, data collection, analysis or dissemination of the work.

**Competing interests:** The authors have declared that no competing interests exist.

patients' care options, professionals' beliefs, locus of control, task planning, and the ability to make decisions based on experience, were indicated as elements that can cause or trigger moral distress. In addition, and relevant to the CoVID-19 pandemic, was use/access to personal protection resources. The social or relational factors were linked to the responsibility for advocating for and communication with patients and families, and professionals own support network. At organisational levels, hierarchy, regulations, support, workload, culture, and resources (staff and equipment) were identified as elements that can affect professionals' moral comfort. Patients' care, morals/beliefs/standards, advocacy role and culture of context were the most referenced elements. Data on cultural differences and diversity were not sufficient to make assumptions. Lack of resources and rapid policy changes have emerged as key triggers related to the pandemic. This suggests that those responsible for policy decisions should be mindful of the potential impact on staff of sudden and top-down change.

## Conclusion

This review indicates that causes and triggers of moral injury are multifactorial and largely influenced by the context and constraints within which professionals work. Moral distress is linked to the duty and responsibility of care, and professionals' disposition to prioritise the wellbeing of patients. If the organisational values and regulations are in contrast with individuals' beliefs, repercussions on professionals' wellbeing and retention are to be expected. Organisational strategies to mitigate against moral distress, or the longer-term sequalae of moral injury, should address the individual, social, and organisational elements identified in this review.

## Introduction

Moral *distress* in Health and Social Care Workers (HSCW) describes the psychological unease that is triggered when a professional identifies the ethically or morally correct action to take in a situation but is unable to perform this due to constraints [1]. Lack of actual or perceived power or agency over the situation, structural or organisational limitations such as low staff ratios, lack of education or time can limit healthcare professionals' actions [1]. First described by Jameton in 1984 [2], the term illustrates the experience of healthcare workers (especially nurses) working within hierarchical structures where they are prevented from acting in a way that aligns with their own ethical standards. Differing approaches and definitions of moral distress, due to a focus on potential causes rather than developing a clear definition of the concept with common consent, has meant slow progress in this research area. In addition, there is a lack of longitudinal data on moral distress [3], which might provide insight into the long-term impact on individuals and organisations. More recently, attempts to reconcile the various definitions and associated terms have concluded that moral distress is the results of three components: moral conflict (two moral principles conflict with one another), moral judgement (evaluation of what is good or bad) and an obstruction that prevents one acting on this judgement [4]. Moral distress has also been associated with burnout (a state of physical and emotional exhaustion) [5], compassion fatigue (stress resulting from exposure to traumatised individual (s)) and moral failure (when moral dilemmas arise from forced choices according to a systemic preference rather than what is morally acceptable) [6–8].

Whilst moral distress can be viewed as situational and sometimes preventable, moral *injury* is the consequence of sustained moral distress (a single episode of great impact may be sufficient) and has been linked to severe mental health issues [1]. Key components include an identifiable, morally correct course of action along with a perception of lack of ability, means, or power to execute that course of action. Such limitations could be due to institutional, interpersonal or regulatory constraints (e.g., care prioritization and resource allocation [9]). The consequences can be emotional, psychological or behavioural, due to internal dissonance (a breach in the individual's moral identity and inner self) [4] and may be characterised by a functional impairment [10]. These facets of long-lasting impact are what separates moral distress from moral injury.

There is a clear distinction between moral injury and post-traumatic stress disorder (PTSD). Moral injury relates specifically to an act (or acts) that violate deeply held morals, whereas PTSD relates to actual of threatened death or serious injury. The consequences can be similar in terms of re-experiencing and avoidance or numbing, but PTSD can also include physiological arousal with escalation of blood pressure and respiratory rate [11]. Moral injury and PTSD are also distinct from burnout which can be a consequence of moral injury but can have other underlying causes unrelated to the morality of a situation [1].

Recent UK data collected by the British Medical Association (BMA), suggests experience of moral distress is common. Of more than 1,900 doctors surveyed, 78.4% reported that moral distress resonates with their experience. Younger, less experienced doctors, and those from minority ethnic groups or with disabilities appear disproportionately affected [1]. Insufficient service staffing is reported as the biggest contributory factor to moral distress. The BMA reported that 53% of respondents cited this [1]. Mental fatigue and a lack of time or inability to provide emotional support and timely treatment were other key factors. Additionally, doctors from minority ethnic backgrounds mentioned inequalities in the ability to speak up in the workplace [1]. Approximately half (51.5%) of respondents to the BMA survey indicated that their experience resonated with moral injury [1].

High levels of moral injury in HSCW (as assessed via self-report questionnaires) have been associated with anxiety, depression and PTSD [12,13] as well as suicidal ideation [12]. Burnout is linked to moral injury via its association with secondary traumatic stress. HSCW reported that listening to first-hand trauma experiences of another is a contributing factor to burnout and moral injury [14]. Moreover, psychological stressors at work such as a poor ethical climate, autocratic leadership, or high emotional demands have been identified as predictors of moral injury [15–18].

It has been demonstrated that sustained levels of moral distress and subsequent moral injury pose significant personal and institutional consequences including burnout, poorer patient care, and challenges with staff recruitment and retention [1,19–22]. The CoVID-19 pandemic provided a context which exacerbated moral injury amongst healthcare workers because it increased existing workforce shortage and disrupted "normal" service delivery. Evidence from many prospective research studies conducted during the pandemic demonstrating the increased incidence of mental health disorders amongst those that worked during the first wave (the initial major increase in cases and deaths associated with CoVID-19 from early 2020) [23,24]. One survey showed that the greatest impact of moral distress was on nurses, as also reported pre-pandemic [25]. Analysing HSCW experiences of moral injury and distress to identify associated triggers and factors may suggest sustainable organisational changes that support workers wellbeing and retention, and reduce the long-term impact of moral distress on their mental health.

## Aims and objectives

The aim of the review was to examine the views and experiences of moral injury/distress in healthcare workers employed across a range of clinical settings. The primary objective was to explore the causes and triggers of moral injury/distress in HSCW, as described in qualitative studies. Secondary objectives were to examine:

- Psychological safety (i.e., feeling able to speak out and/or seek help) and whether this has any impact on mental wellbeing and/or is associated with moral injury/distress in HSCW.

- Whether diversity/cultural differences were present among HSCW's influence the experience of moral injury/distress, or consequences of it.

- The effect (if any) of major events/disasters (e.g., pandemics, low probability/high impact events) on the experience of moral injury/distress of HSCW.

- Insights into preventative treatment strategies and/or interventions as described by HSCW.

## Methods

This systematic review was conducted and reported according to recommendations from Cochrane [26], guidelines on conducting systematic qualitative views [27,28] and in alignment with Preferred Reporting Items for Systematic Reviews and Meta-Analyses (PRISMA) guidelines [29,30] (see S1 File).

### Search strategy

The full updated protocol including search strategy and terms is in S2 File and the initial review was registered on Prospero [31]. Keywords and search terms were identified using PICO(s):

P = Participants (HSCW)

I = Intervention—no specific intervention will be measured in this qualitative review

C = Context (Experiencing of moral injury/distress)

O = Outcome (experiences and views)

S = Study design (qualitative studies)

Three searches were conducted on 2 main databases: CENTRAL and PubMed (including MEDLINE) between June 2021 and January 2024. Three specialist databases were searched (Scopus, CINAHL and PsycArticles), alongside medRxiv for pre-prints, study registries (e.g., ClinicalTrials.gov, ISRCTN registry), and systematic review reference lists. The "cited by" function in Google Scholar was used for papers selected at full-text screening. In accordance with the protocol, search terms combined relevant key words and MeSH terms related to moral injury, participants and study design. Words were adapted according to the database. The full search strings are reported in the published protocol.

### Inclusion and exclusion

#### Inclusion.

➢ Qualitative studies (interviews and focus groups, open question survey/questionnaire).

➢ Healthcare and allied multidisciplinary workers

Including nurses, physicians, consultants, GPs, paramedics/EMTs, occupational therapists, physical therapists, psychotherapists, psychologists, psychiatrists, counsellors, anaesthetists, epidemiologists, nutritionists, patient care teams, audiologists, caregivers, case managers, coroners and medical examiners, optometrists and anyone identified as health personnel, allied health personnel, hospital workers or health and social care workers.

- All healthcare settings including primary and secondary care, public or private practice.

➢ Moral injury/distress cited within the primary aim of the study.

➢ Written in English or Italian (due to the spoken language of reviewers).

➢ Conducted in North America (Canada & USA), Europe (EU, candidate EU and Schengen Area), New Zealand & Australia.

➢ Articles published after 2011, 10 years before the beginning of this review (see later for details).

**Exclusion.**

➢ Review article, book, editorial or similar non-original research study.

➢ Solely quantitative in design or mixed methods.

➢ Non-healthcare workers (e.g., military personnel, other frontline workers such as police, fire and rescue, hospital management and students of healthcare professions).

➢ Absence of main outcomes or aims related to moral injury/distress e.g., focus on general wellbeing or mental health.

➢ Studies outside geographical remit.

**Study selection and data extraction.** At least two of eight reviewers (EB, AS, AW, JM, MW, HMW, DT, AC) from a combination of research and clinical backgrounds (psychologists, critical care consultants and academic researchers) across two institutions screened all title, abstracts, and full text articles. One was a native Italian speaker which allowed the inclusion of papers from this region. Where disagreement occurred, consensus on eligibility was discussed with a third reviewer.

The literature review was conducted using a research collaboration platform: Rayyan [32], search results were uploaded and de-duplicated. The PRISMA flow diagram in Fig 1 details the study selection and reasons for exclusion. Articles were initially blind assessed independently by a minimum of 2 reviewers and allocated an include, exclude and maybe label using the free version of the web-based programme.

The study selection criteria were tested on an initial set of 5 abstracts in an all-team review. The remaining abstracts were assessed by 2 reviewers (EB & AS), with a third reviewer to resolve any disagreement around inclusion (AC or AW). Referral on to the whole team occurred where one or more reviewers felt an article had been incorrectly excluded. Articles categorised by reviewers as maybe, were included for the title/abstract screening stage, as the full text was required to clarify their inclusion. There was 81% agreement prior to conflict resolution and consensus was reached via whole-team discussion.

Any systematic reviews identified at the screening stage were flagged for further exploration to mitigate against omission. Two reviews were identified via hand searching and assessed by one reviewer (MW) and 2 studies added to the review.

Following screening, 7 of the 8 reviewers independently extracted data and assessed quality and risk of bias using the CASP guidance, a tool to systematically assess articles quality [33].

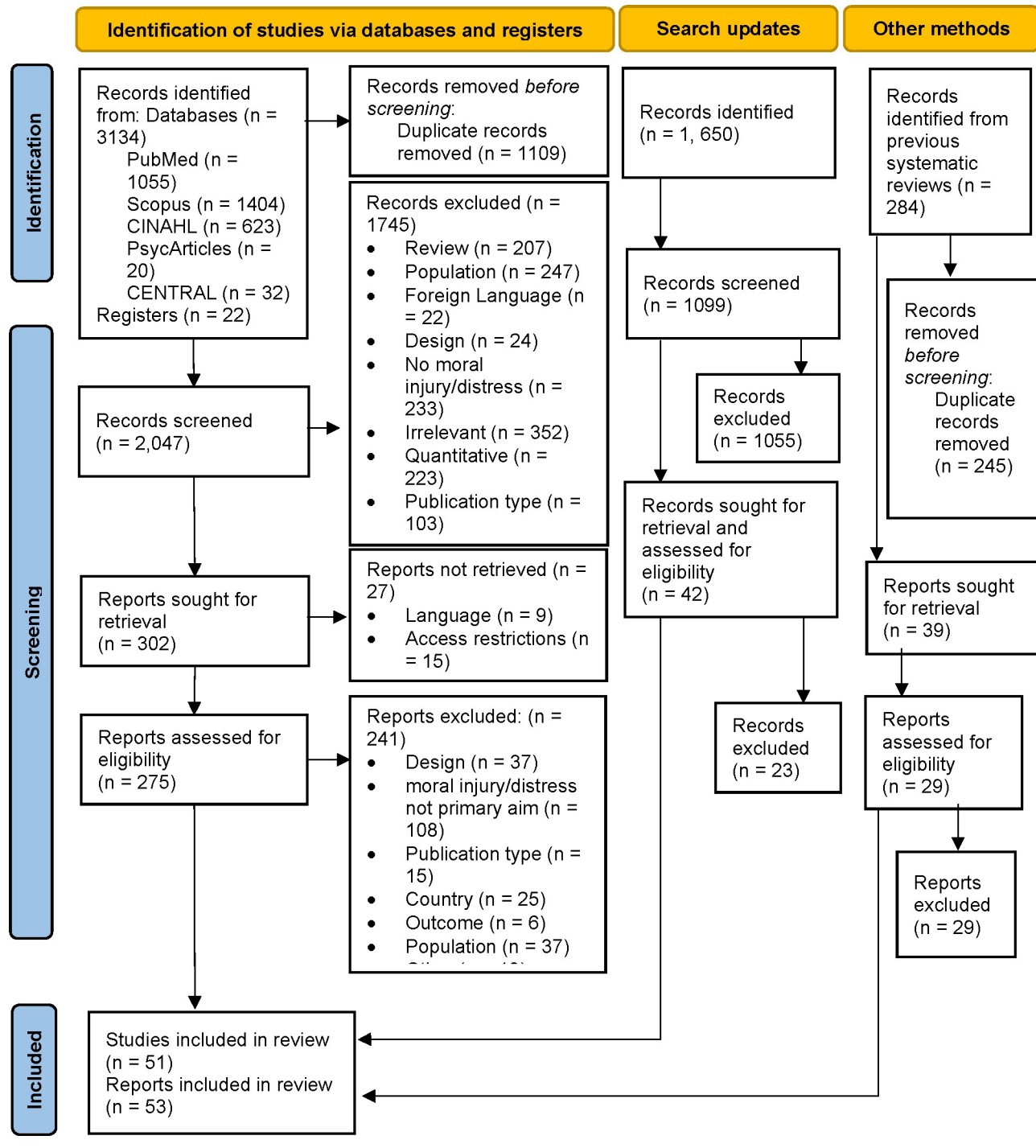

**Fig 1. PRISMA diagram of the searches, screening and included studies.**

All data extraction was checked by at least one other of three reviewers (EB, AC and DT), to ensure consistency.

The data collected included basic study information (e.g., citation, aims, country), methodological data (e.g., study design and sample), results summary and discussion points (e.g., limitations) to provide context to the reviews and facilitate the findings summary (see S3 File).

Any additional sources of results such as supplementary materials were extracted in full and analysed as detailed below.

### Data analysis and synthesis of studies verbatim quotes

Thematic analysis has been identified as an appropriate method for synthesis of data in systematic reviews and has been applied successfully to several studies [34,35]. Thematic analysis was conducted on the verbatim quotes as part of a three-stage process; line by line coding of text, generation of descriptive themes and the development of analytical themes [36]. For this reason, two articles not providing verbatim quotes regarding causes and triggers were excluded after being shortlisted.

Verbatim findings were entered into NVIVO [37] and then coded by individual reviewers line-by-line into codes, in order to begin organising the data. Coding was annotated with reviewers' comments but at this point there was no structuring of codes or further categorisation. Coding was undertaken using an inductive approach within the context of the research question. Since multiple researchers were coding, codes underwent a continuous review process to assess patterns across articles, this iterative step preceded the development of analytical themes in which coding categories were generated. Codes describing patterns in the data and themes were built around proposition statements and explanations with implications for moral distress/injury. By grouping codes into categories, themes began to emerge for which definitions were drafted to capture overarching meaning. Using only verbatim quotes, we stepped beyond the primary studies to develop explanations for the triggers, factors and mitigators of moral distress. Analytical themes were generated by 'going beyond' the summary and conclusions of the original studies' [34] and using a small group of reviewers including clinical staff at each stage of coding, facilitated an ongoing reality check on interpretation and minimised any impact of personal bias [28].

### Risk of bias/quality assessment

Quality of selected studies was assessed using the CASP tool for qualitative studies [33] to ensure systematic assessment of all important factors and ensure consistency in scoring. Reviewers worked in pairs to independently assess and then compare scores across ten domains of quality including design, recruitment, ethics, and value of the research in terms of contribution to existing knowledge. A quality score was assigned according to how well the study met each criterion; 0 = not met at all, 1 = somewhat achieved, 2 = achieved. The overall score was on a range between 0 and 20; scores <10 were rated 'poor', scores between 10–15 'fair', scores of 16–18 'good' and scores of 19–20 'excellent'. To address any issues or scoring differences, the reviewers met (online, if required) to discuss their decisions and to consolidate the final quality appraisal decisions. The agreed summary of quality assessment is shown in Table 1.

## Results

### Study characteristics

A total of 3418 articles were identified via databases and registers and a further 1650 articles were identified from search updates. There were 1660 duplicate articles which were removed. Overall, 3146 studies were screened and a final list of 51 articles (49 studies) were included in the full review and data extraction. Nine (17.3%) were assessed as excellent, 26 (50%) were of good quality, while 17 (32.7%) were assessed as fair. Articles published from 2011 onwards were included to reflect recent organisational changes and focus on staff wellbeing. Also, it is

**Table 1. Quality assessment of the included studies using the CASP criteria for qualitative designs.**

| | Aims | Qualitative methodology | Design | Recruitment | Data collection | Researcher reflexivity | Ethics | Analysis rigor | Findings statement | Value | Overall |
|---|---|---|---|---|---|---|---|---|---|---|---|
| Ahokas et al., 2023 [38] | | | | | | | | | | | Good |
| Arends et al. 2022 [39] | | | | | | | | | | | Good |
| Beck et al., 2022 [40] | | | | | | | | | | | Good |
| Bernhofer & Sorrell, 2015 [41] | | | | | | | | | | | Fair |
| Bourne & Epstein, 2021 [42] | | | | | | | | | | | Good |
| Brasi et al. 2021 [43] | | | | | | | | | | | Excellent |
| Clark et al. 2022 [44] | | | | | | | | | | | Fair |
| Demir et al., 2023 [45] | | | | | | | | | | | Fair |
| Denham et al., 2023 [46] | | | | | | | | | | | Good |
| Deschene et al 2023 [47] | | | | | | | | | | | Excellent |
| Fantus et al., 2023 [48] | | | | | | | | | | | Good |
| Forozeiya et al. 2019 [49] | | | | | | | | | | | Good |
| Foster, 2021 [50] | | | | | | | | | | | Good |
| French et al. 2021 [51] | | | | | | | | | | | Good |
| Gagnon & Kunyk, 2021 [52] | | | | | | | | | | | Excellent |
| Gherman et al. 2022 [53] | | | | | | | | | | | Excellent |
| Hancock et al. 2020 [54] | | | | | | | | | | | Fair |
| Helmers et al. 2020 [55] | | | | | | | | | | | Fair |
| Henrich et al. 2016 [56] | | | | | | | | | | | Good |
| Howard et al., 2023 [57] | | | | | | | | | | | Good |
| Jansen et al. 2020 [58] | | | | | | | | | | | Good |
| Jansen et al., 2022a [59] | | | | | | | | | | | Good |
| Jansen et al., 2022b [60] | | | | | | | | | | | Excellent |
| Jodee et al., 2023 [61] | | | | | | | | | | | Fair |
| Kielman et al., 2023 [62] | | | | | | | | | | | Fair |
| Koonce & Hyrkas 2023 [63] | | | | | | | | | | | Excellent |
| Kreh et al. 2021 [64] | | | | | | | | | | | Good |
| Lamiani et al. 2021 [65] | | | | | | | | | | | Excellent |
| Liberati et al. 2021 [66] | | | | | | | | | | | Excellent |
| Lovato & Cunico, 2012 [67] | | | | | | | | | | | Good |
| Matthews & Williamson, 2016 [68] | | | | | | | | | | | Good |
| McCracken et al. 2021 [69] | | | | | | | | | | | Good |
| Molinaro, Polzer et al., 2023 [70] | | | | | | | | | | | Good |
| Molinaro, Shen et al., 2023 [71] | | | | | | | | | | | Fair |
| Morley et al. 2020 [72] | | | | | | | | | | | Good |
| Morley et al. 2022 [73] | | | | | | | | | | | Good |
| Musto & Schreiber, 2012 [74] | | | | | | | | | | | Fair |
| Pye, 2013 [75] | | | | | | | | | | | Fair |
| Ritchie et al., 2018 [75] | | | | | | | | | | | Fair |
| Robinson & Stinson, 2016 [76] | | | | | | | | | | | Fair |
| Scott et al., 2023 [77] | | | | | | | | | | | Good |
| Silverman et al. 2021 [78] | | | | | | | | | | | Excellent |
| Smith et al., 2023 [79] | | | | | | | | | | | fair |
| St Ledger et al. 2021 [80,81] | | | | | | | | | | | Fair |
| Sukhera et al. 2021 [82] | | | | | | | | | | | Good |
| Thomas et al. 2016 [83] | | | | | | | | | | | Fair |
| Thorne et al. 2018 [84] | | | | | | | | | | | Good |
| Trachtenberg et al. 2022 [85] | | | | | | | | | | | Fair |
| Villa et al. 2021 [86] | | | | | | | | | | | Good |
| Walt et al. 2022 [87] | | | | | | | | | | | Good |
| Weiste et al., 2023 [88] | | | | | | | | | | | Good |

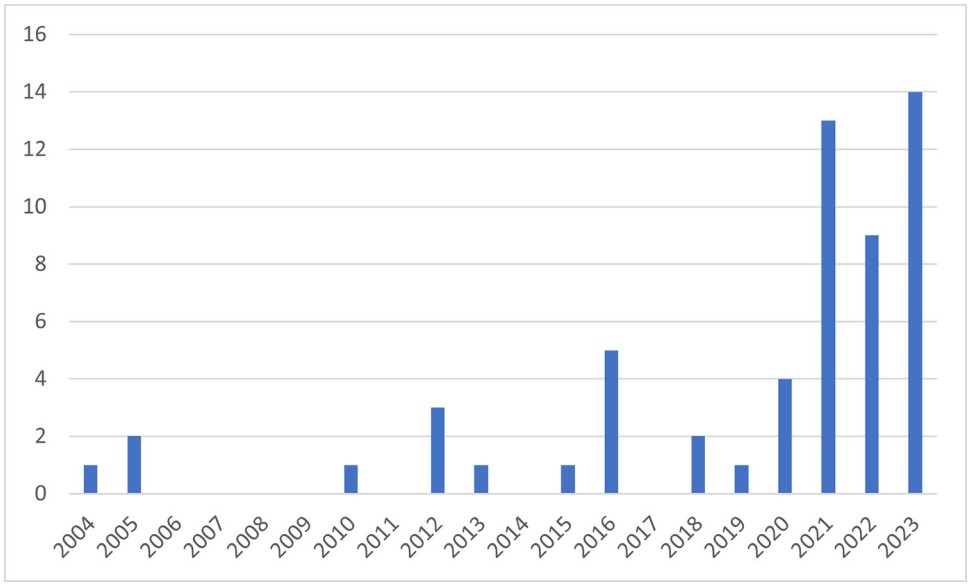

**Fig 2. Number of qualitative articles on moral distress/injury by year of publication.**

noted that published qualitative evidence on moral distress and injury before 2011 was scarce (Fig 2). The 51 articles (summarised in Table 2) included studies conducted in different settings including oncology, mental health, emergency, intensive care and end of life care settings; and involving healthcare professionals working with both younger and adult patients. Of those studies that reported sample size, approximately 1188 participants were included and of the total participants (where breakdown by profession was reported) the majority were nurses (~45.4%). Other professionals included doctors (of varying levels), healthcare assistants, psychologists, social workers and physiotherapists.

## Reasons for and factors that influence moral injury/distress in healthcare workers

Exploring experience of moral distress and moral injury as reported in fifty-one qualitative studies involving health care professionals, led to an understanding of the reasons for and factors that influence moral distress and moral injury as the primary objective. Causes and triggers of moral distress are presented in Table 3 and Fig 3. Contributory factors with reference to diversity, culture, and exceptional circumstances are presented subsequently. Causes and triggers codes, extrapolated from verbatim quotes, were identified as fitting within three categories: individual, organisational, and social or relational. Table 3 shows the categories and the subcategories of codes by frequency. Frequency refers to the number of articles that report quotes coded to the respective subcategory. The last column shows how many quotations referred to each subcategory. Patient's care issues, patients and family's advocacy and communication, and context culture were the subcategories that participants most referred to, followed by morals, beliefs and standards, and regulations and procedures. Fig 3 shows how the categories and subcategories overlap as some causes and triggers relate to more than one domain.

Table 4 shows a sample of the quotations from participants of primary studies to illustrate the subcategories and the aggregated themes describing triggers and factors.

**Table 2. Summary of study characteristics of the included studies.**

| Authors, Year | Location, Sample | Pandemic Perspective | Recruitment strategy | Data collection | Theoretical approach |
|---|---|---|---|---|---|
| Ahokas et al., 2023 [38] | Finland, older care, 8 care leaders | During pandemic | Purposive sampling | Semi-structured interviews | Content analysis |
| Arends et al., 2022 [39] | Netherlands, Mixed hospital wards, 23 nurses | During pandemic | Snowball sampling | Semi-structured interviews | Thematic analysis |
| Beck et al., 2022 [40] | USA, Teaching hospitals, 40 doctors | N/A | Purposeful sampling | Semi-structured telephone interviews | Inductive approach to thematic analysis with constructivist paradigm |
| Bernhofer & Sorrell, 2015 [41] | USA, Surgical & emergency department, 48 nurses | N/A | Invitation | In-depth interviews | Grounded theory |
| Bourne & Epstein, 2021 [42] | USA, Family medicine, 12 primary doctors & nurse practitioners | N/A | *Not reported* | Semi-structured interviews (Face-to-face or telephone) | Conventional content analysis |
| Brasi, et al., 2021 [43] | Italy, Oncology & haematology, 40 nurses | N/A | Snowball sampling | In-depth interviews | IPA |
| Clark et al., 2022 [44] | USA, Emergency department, 15 nurses | N/A | Purposeful sampling | Semi-structured interviews | Theoretical qualitative content analysis |
| Demir et al., 2023 [45] | Sweden, 2 hospitals, 12 nurses | N/A | Purposive sampling | Semi-structured interviews | Inductive content analysis and Framework. Critical reflective approach. |
| Denham et al., 2023 [46] | UK, various, 20 mixed | During pandemic | Previous study subsample | Semi-structured interviews | Critical Realist, Reflexive thematic analysis. |
| Deschenes et al., 2023 [47] | Canada, Paediatric ICU, 10 nurses | During pandemic | Purposive sampling | Semi-structured interviews | Qualitative description |
| Fantus et al., 2023 [48] | USA, mixed, 43 healthcare social workers | During pandemic | Purposive, snowball sampling | Semi-structured interviews | Directed content analysis |
| Forozeiya et al., 2019 [49] | Canada, ICU, 7 nurses | N/A | Purposeful sampling | Semi-structured interviews | Thorne's approach to interpretive description |
| Foster et al., 2022 [50] | Australia, Private midwifery, 14 midwives | N/A | Purposeful sampling | Semi-structured online interviews | Thematic analysis/ naturalistic enquiry |
| French et al., 2021 [51] | UK, General practice, 16 mixed | N/A | Convenience sampling | In-depth interviews | Critical realist approach |
| Gagnon & Kunyk, 2021 [52] | Canada Paediatric ICU, 7 nurses | N/A | Invitational sampling | Unstructured interviews | Narrative inquiry |
| Gherman et al., 2022 [53] | Romania, Mixed, 25 nurses | N/A | Snowball sampling | Episodic semi-structured interviews | Constructivist thematic analysis |
| Hancock et al., 2020 [54] | Canada, ICU care, 35 mixed | N/A | Convenience sampling | Focus groups | Thematic analysis |
| Helmers et al., 2020 [55] | Canada, Paediatric ICU, 17 nurses | N/A | Convenience sampling | Focus groups/ semi-structured interviews | Sequential two-phase iterative approach |
| Henrich et al., 2016 [56] | Canada, ICU's, 56 mixed | N/A | Convenience sampling | Focus groups/ telephone interviews | *Not reported* |
| Howard et al., 2023 [57] | USA, various, 18 occupational therapists | During pandemic | Stratified purposive sampling | Semi-structured interviews | Hermeneutical phenomenology |
| Jansen et al., 2020 [58] | Norway, Acute psychiatry, 16 nurses | N/A | Purposeful sampling | In-depth interviews | Thematic analysis |
| Jansen et al., 2022a [59] | Norway, Psychiatric service, 30 nurses | N/A | Purposeful sampling | In-depth interviews [16] & focus groups [14] | Thematic analysis |

*(Continued)*

**Table 2.** (Continued)

| Authors, Year | Location, Sample | Pandemic Perspective | Recruitment strategy | Data collection | Theoretical approach |
|---|---|---|---|---|---|
| Jansen et al., 2022b [60] | Norway, Acute psychiatric hospitals, 44 nurses | N/A | Purposeful sampling | Focus groups [14] & interviews [30] | Thematic analysis |
| Joolaee et al., 2023 [61] | Canada, ICU, 24 mixed | N/A | Convenience sampling | Survey questions | Participatory action research approach, Content analysis |
| Keilman et al., 2023 [62] | USA, community hospital, 20 nurses | Pre-COVID data | Convenience sampling | Semi-structured interviews | Unclear |
| Koonce et al., 2023 [63] | USA, tertiary care teaching hospital, 9 pulmonary care nurses | During pandemic | Purposive sampling | Semi-structured interviews | Descriptive phenomenology |
| Kreh et al., 2021 [64] | Italy & Austria, 13 doctors, nurses & psychologists | N/A | Invitation | Focus groups, mixed method interviews | Grounded theory |
| Lamiani et al., 2021 [65] | Italy, Emergency department & ICU, 15 doctors | During pandemic | Snowball sampling | Semi-structured in-depth interviews | Grounded theory |
| Liberati et al., 2021 [66] | UK, Community mental health services, 35 mixed | During pandemic | Purposeful strategy/ snowball sampling | Semi-structured in-depth interviews | Constant comparative method |
| Lovato & Cunico, 2012 [67] | Italy, Adult medicine, 40 nurses | N/A | Purposeful sampling | Focus groups/ diaries | Thematic analysis |
| Matthews & Williamson, 2016 [68] | UK, Secure mental health hospital, 10 healthcare assistants | N/A | Convenience sampling | Diaries/ semi-structured interviews | IPA |
| McCracken et al., 2021 [69] | USA, Oncology, 32 mixed | N/A | Convenience sampling | Semi-structured focus group | Content analysis |
| Molinaro, Polzer et al., 2023 [70] | Canada, mixed, 9 paediatric oncology nurses | N/A | Purposive sampling | semi-structured interviews | Grounded feminist approach, |
| Molinaro, Shen, et al., 2023 [71] | Canada, various primary care, 20 GPs | Discussed | Purposive sampling | Narrative interviews | Critical narrative approach |
| Morley et al., 2020 [6] | UK, ICU, 30 nurses | N/A | Purposeful sampling | Phenomenological interviews | Feminist interpretive phenomenology |
| Morley et al., 2022 [72] | UK, ICU, 30 nurses | N/A | Purposeful sampling | Phenomenological interviews | Feminist interpretive phenomenology |
| Musto & Schreiber, 2012 [73] | Canada, Inpatient & outpatient mental health, 12 nurses | N/A | *Not reported* | Semi-structured in-depth interviews | Glaserian grounded theory |
| Pye, 2013 [74] | UK, Paediatric oncology, 4 nurses, 4 doctors | N/A | Self-selected sample | Open-ended semi-structured discussion from hypothetical scenario | Colaizzi's [1978] descriptive framework & Riley's (1996) method of qualitative analysis |
| Ritchie et al., 2018 [75] | Canada, Community care, 6 nurses | N/A | Purposeful sampling | Semi-structured interviews | Interpretive description |
| Robinson & Stinson, 2016 [76] | USA, Emergency department, 8 nurses | N/A | Convenience sampling | Structured in-depth interviews | Phenomenological approach |
| Scott et al., 2023 [77] | UK, hospital, 17 nurses | During pandemic | Unclear | Semi-structured interviews | Constructivist paradigm |
| Silverman et al., 2021 [78] | USA, Teaching medical centres, 31 nurses | During pandemic | Purposeful sampling | Focus groups and in-depth interviews | Thematic framework, apriori categories; individual, relational, organisational & systematic |
| Smith et al., 2023 [79] | Canada, various, 78 mixed (13 nurses) | During pandemic | Purposive sampling | Semi-structured interviews and focus groups | Framework analysis using gender-based analysis. |

(*Continued*)

**Table 2.** (Continued)

| Authors, Year | Location, Sample | Pandemic Perspective | Recruitment strategy | Data collection | Theoretical approach |
|---|---|---|---|---|---|
| St Ledger et al., 2021 [80,81] | UK, ICU, 18 physicians | N/A | Purposeful sampling | Interviews | Thematic analysis |
| Sukhera et al., 2021 [82] | Canada, Mixed, 22 junior doctors | During pandemic | Convenience sampling | In-depth interviews | Constructivist grounded theory |
| Thomas et al., 2016 [83] | USA, PICU, 25 mixed | N/A | Convenience sampling | Semi-structured interveiws | Thematic analysis |
| Thorne et al., 2018 [84] | Canada, Neonatal ICU, 28 mixed | N/A | Convenience, snowball sampling | Semi-structured interviews | Interpretive description |
| Trachtenberg et al., 2022 [85] | USA, Teaching hospital, 16 nurses & 4 therapists | N/A | Purposeful sampling | In-depth semi-structured interviews | Inductive & deductive approach |
| Villa et al., 2021 [86] | Italy, Elderly care, 13 mixed | N/A | Purposeful sampling | Interviews | Grounded theory |
| Walt et al., 2022 [87] | USA, Emergency department, 21 mixed | N/A | Convenience sampling | Semi-structured in-depth interviews | Thematic/narrative analysis |
| Weiste et al., 2023 [88] | Finland, older care, 38 unit supervisors and care workers | N/A | Purposive sampling | Focus groups | Inductive approach, Conversational and discursive analysis |

(ICU: Intensive Care Unit; UK: United Kingdom; USA: United States of America; IPA: interpretative phenomenological analysis; Pre-COVID: data collected prior to COVID but analysed and published post-COVID, During-COVID: data collected during pandemic; Discussed: COVID pandemic is discussed but role in data collection is unclear; N/A: no described role of pandemic).

## Description of individual causes and triggers categories

<u>Individual causes and triggers</u>, as linked to the personal domain rather than having a social or organisational origin, can include concerns related to patients' care, events or circumstances that conflict with one's own morals and beliefs, control and responsibilities, time and planning, level of seniority, and exposure to personal risks:

**Table 3. Categories and subcategories of causes and triggers of moral distress and moral injury.**

| Categories | Subcategories | Number of articles (frequency) | References to subcategories within the quotations |
|---|---|---|---|
| Individual | Patient's care issues | 37 | 171 |
| | Morals, beliefs, and standards | 30 | 72 |
| | Control and power, and responsibility | 29 | 51 |
| | Time and priorities | 28 | 55 |
| | Work experience, seniority, and training | 15 | 15 |
| | Personal risk (especially during pandemic) | 14 | 37 |
| Social or relational | Patients and family's advocacy and communication | 43 | 108 |
| | Support network, personal and peer-to-peer | 11 | 21 |
| Organisational | Hierarchy and seniority | 22 | 50 |
| | Workload and roles | 12 | 23 |
| | Trust and support | 26 | 40 |
| | Regulations and procedures | 28 | 65 |
| | Context culture | 43 | 77 |
| | Resources: equipment, staffing | 30 | 50 |

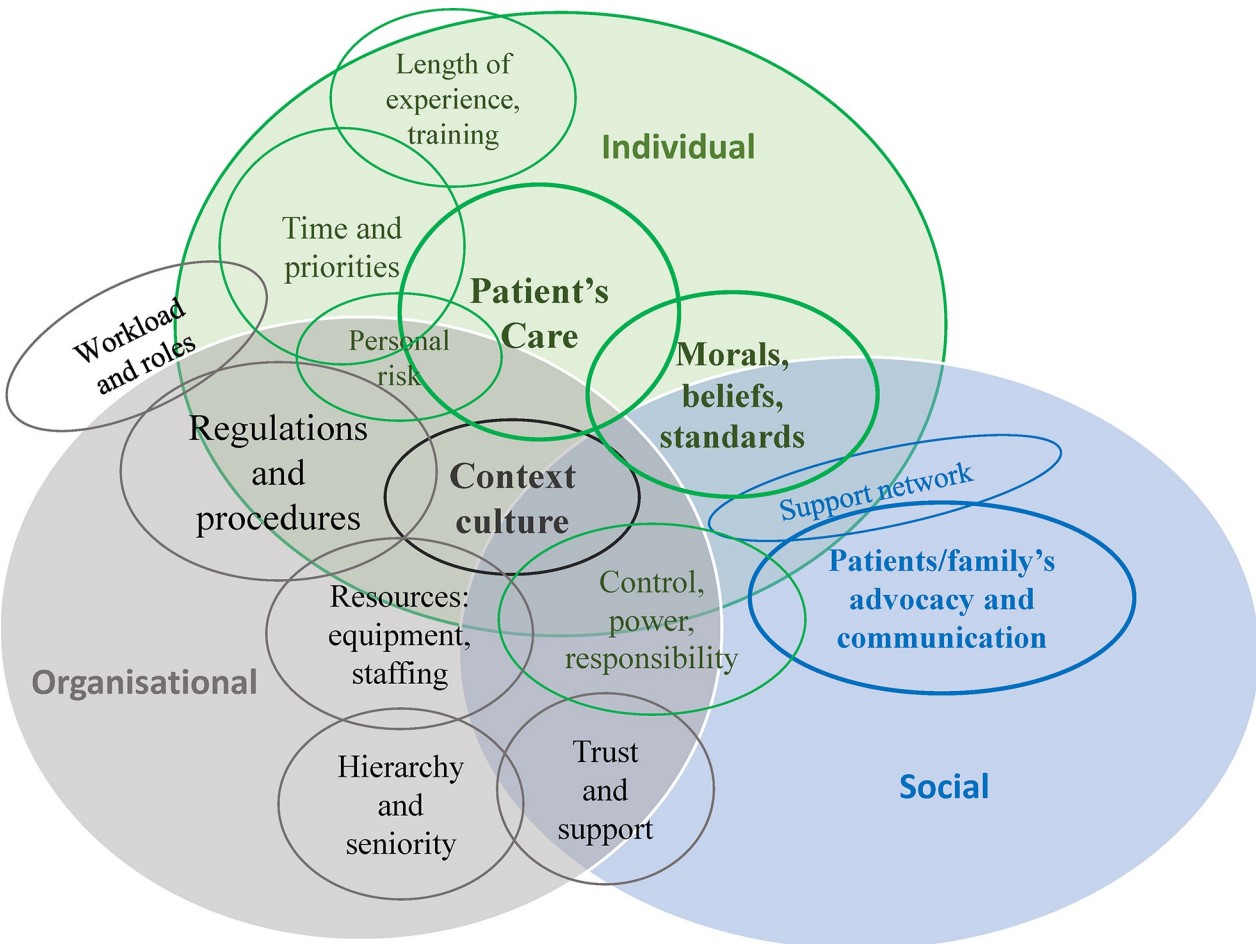

**Fig 3. Categories related to individual, organisation, and social causes and triggers of moral distress and moral injury.**

*Patients' care*—Participants described potential causes of distress linked to patients' treatment. For example, when causing pain, especially in children, cases of care perceived as futile, or situations when care professionals were not able to act in accordance with patient wishes. Triggers of internal conflict and moral reflections were linked to feelings of empathy, described as caring about the patients' wellbeing and putting oneself in the patients' position. Treatment of patients that failed to improve their quality of life or the restrictions patients' face linked to insurance constraints, as well as the lack of empathy shown by colleagues were reported by professionals covering different roles and levels of seniority [e.g., 56,69,72,74,78,84,85,87].

*Morals, beliefs, and standards*—When participants were unable to act in line with own morals, beliefs, and standards, and they were unable "doing things right" [66] or there was uncertainty around best treatment choices and related outcomes, this was a cause of distress [e.g., 6,50,76,78]. This happens when actions and choices relating to clinical care are in opposition to professionals' moral judgement *pertaining to others' welfare, rights, fairness, or justice*, also indicated as moral reasoning (p3. [89]).

**Control and responsibilities.** In studies focussing on the experience of nurses, but also mentioned by other professionals, issues around control and power over patient care were commonly described as a source of moral distress. Delegation of duties varies according to

**Table 4.** Examples of quotations linked to the primary objective from original papers.

| Review themes | Sub-category | Quotations from primary articles that relate to the sub-category and themes |
|---|---|---|
| Be in someone else's shoes | Patient's care issues | *"Moreover, the loneliness experienced by patients at the end of life causes morally distressing feelings in nurses."* De Brasi et al [43]<br>*"The same patient came in over and over. It was clear he was going to die. It was hard for me. Finally, I thought enough is enough and part of me feels guilty for feeling that way."* Robinson & Stinson [76] |
| | Morals, beliefs, and standards | *"I do struggle with this whole business about repetitive chemotherapy that is not curative, I really do. I always have done, but the natural law meant that this child would have died long ago and therefore whatever we are doing beyond that is playing God in a sense and doing unnatural things that are often unpleasant for the child."* Pye [74]<br>*"I feel like I'm compromising my morals and ethics and values on a fairly daily basis at the in the environment that I work in."* Foster [50] |
| | Patients and family's advocacy and communication | *"...nobody was just telling her the truth of her situation, and she was begging me to tell her the truth of her situation... I told her it was bad, that's all I said. I said 'Yes, you are right, it's bad.' Hell, yes! There was no way that I was going to lie to her."* Koonce & Hrykas [63]<br>*"The patient told me in confidence, 'I'm done. I do not want this anymore.' But then, their family comes in, and they, you know, put on a brave face for them. And the family wants everything done... so the patient will do it for their family. That's one of the toughest struggles."* McCracken [69]<br>*"I find sometimes moral distress issues come from the fact that the onus of the decision of the kids sometimes is placed on the parents. ... We know what the obvious [decision] should be, but we give it to the parents and the burden of that sometimes causes prolongation [of care] and the distress that we see—and I've seen a lot."* Deschenes [47] |
| Experience leads to autonomy and increasing responsibility | Control and power, and responsibility | *"I don't have as much information to make decisions so I'm questioning my decision-making more thoroughly. I'm frightened of making the wrong decision when I'm deciding whether somebody gets a service or doesn't get a service. That's quite problematic."* Liberati [66]<br>*"But certainly, the stress of knowing that baby has died, and I've contributed to that . . . It takes a piece of your soul."* Foster [50]<br>*"It's hard to be in the middle of the hierarchy. . . giving a lot of responsibilities but no power to make meaningful change."* McCraken [69] |
| | Work experience, seniority, and training | *"When you're a senior member of the team you've got to sort of keep up appearances. Even when you're not really all right. If you start to crumble then that's not good for the rest of the team."* Scott [77]<br>*"I think it's hard when, you know, your charge nurse has two years of experience and they're managing a floor of, you know, 10, 15 nurses that have less than a year and a half of experience and it's almost like the blind leading the blind [...] there's been a LOT of BIG mistakes, on that unit."* Molinaro [70] |
| What comes first? Me, the patients, the organisation | Time and priorities | *"... you would need to have sufficient time for your patients, that's the biggest [problem linked to moral distress] ... It's reflected onto the patients ... even if the caregivers are as nice and capable as you like they don't have time to perform the work properly. The physical [work] but the mental [work] becomes probably left half-done".* Ahokas [38]<br>*"Moral distress is about the 'to-do list' constantly replenishing. I must deprioritize something important."* Demir [45] |
| | Personal risk (especially during pandemic) | *"When you're trying to deal with CoVID and you've got additional stressors like a shortage of gowns and masks which are out of date it isn't good."* Scott [77]<br>*"It is a background low level of stress, not to mention the terror of getting sick and dying. . . .There was always that fear of what happens if I get sick, and what happens if I bring it home."* Howard [57] |
| Struggling to meet demand | Resources: equipment, staffing | *"During the very first days. . . I heard only the sound of sirens, the beds seemed to be never enough, the patients were going very bad, they dropped like flies. Those days I said crying: «It will be a massacre!» I really had to say that I was suffering for humankind."* Lamiani [65]<br>*"When you're trying to deal with CoVID and you've got additional stressors like a shortage of gowns and masks which are out of date it isn't good".* Scott [77] |
| | Workload and roles | *"[You] can sometimes feel like you're juggling about 10 balls in the air. Because you're dealing with so many people. [and] we're all over the zone. I'm so afraid of dropping a ball. To someone's detriment."* Ritchie [75]<br>*"Our director that's a nurse is more cut and dry, of 'this is going to be the outcome. The family just needs to get on board,' where, as a social worker, we try to see where the families are at and try to help them get to whatever the outcome will be. That has been a conflict."* Fantus [48] |

*(Continued)*

**Table 4.** (Continued)

| Review themes | Sub-category | Quotations from primary articles that relate to the sub-category and themes |
|---|---|---|
| Counting on others | Support network, personal and peer-to-peer | *"I understand that dealing with certain topics, issues, especially about death, with people who don't live it. . . it's difficult and most of the time they don't understand you. Those few times I talked about it, I didn't feel understood."* De Brasi [43]<br>*I take it home. It affects my family life, it affects my relationships, it affects my patients, and my relationships with my peers."* Robinson [76]<br>*"When your home life is out of kilter and you don't have your support people, you don't have things to put back into your emotional bank, then you start to run on empty and then you take Benadryl and wine and hope that everything goes away"* Smith [79]<br>*"When you have someone you can't trust or to talk to or ask for help, even if it's to grab you something or talk to about some sort of issue that happened, you really feel constrained. You feel your world has definitely gotten a lot smaller."* Deschene [47] |
| | Trust and support | *"You really rely on members of the team and when you feel like you can't, that leaves you feeling pissed off at best and vulnerable and unsafe at worst!"* Matthews & Williamson [68]<br>*"The stress of feeling like I had to do everything on my own and looking after 9 different patients, with no support and ultimately the panic of finding someone with a ligature was pretty stressful! I was mostly angry for being left in that situation and felt so unsupported."* Matthews &Williamson [68] |
| Top-down approach | Hierarchy and seniority | *"It's frustrating because I have had physicians laugh at me when I have asked for a palliative care consult. And, you know, the patient passed away within the month. I feel like they were almost cheated."* McCracken [69]<br>*". . .they (upper management) are sending us to take assignments in other units, and being used as the float pool for the hospital. The staff do not choose to work in those areas; they also feel unsafe and distressed about this policy, but their (management) response is that a nurse is a nurse."* Joolaee [61] |
| | Regulations and procedures | *". . .I am having to choose between getting my green check mark, which eventually translates into payment, either for myself or the hospital, and choosing to do the right thing for the patient. . .I think that is a very dangerous position for a provider to be put in."* Bourne & Epstein [42]<br>*"That is, one of the most annoying parts of our job that we can't do a good job, we KNOW what we're supposed to do but we can't do it the way we want to."* Molinaro et al [70] |
| | Context culture | *"It was this kind of 'get on with it otherwise we'll find someone else to do your job' and that's really gutting really when you've worked in healthcare for most of your life and you were trying to do the best that you could [. . .] it's put me off even working for the NHS. . ."* Denham [46]<br>*"You're working a lot with people who have very different values . . . It doesn't make them wrong or you right, but it is a constant conflict and the constant weighing up . . . and if you not working in an environment that is similar, then every single day is filled with moral dilemmas."* Foster [50] |

role and responsibility, and control over patients' care may not sit with the care giver. This could be difficult to accept for the latter [e.g., 63,69,72–74].

**Time and priorities.** Lack of time and having to prioritise among many patients in need and other tasks is another potential cause of increased moral distress. Participants mentioned that poor planning affects time for patient's care and decision making, which could sit in the organisational category, but could also be related to the time professionals need relevant to their experience and knowledge. HCSWs would like to offer more time to patients and families, especially for providing reassurance and comfort, but it is not always possible because of the need to prioritise other tasks [e.g., 69,72,78,85].

**Work experience, seniority, and training.** The length of work experience, seniority, and training have been often named when discussing potential causes of distress. Staff who lack experience, because they are new to the role or confronted with duties they are not familiar with, may feel overwhelmed and inadequate. Training is not always offered at the right time because of shortages of resources and staff to cover training time, so there is the need for peer-to-peer support that can increase the burden on senior staff [e.g., 6,67,72,78].

### Personal risks

Exposure to personal risks have been mentioned often in articles related to the CoVID-19 pandemic [e.g., 72,78,85]. Recollections of stressful situations that led to experience of moral distress and injury were numerous and include shortages of protective equipment. They will be mentioned later in the exceptional circumstances section.

### Social causes and triggers category

The second category of triggers includes social or relational factors: advocacy for and communication with patients and their families, and professionals' own circle of support, including peer-to-peer relations.

**Patients and family's advocacy and communication.**   Professionals' moral values are affected when there is a mismatch between their own understanding of patients' conditions and what patients are told or offered, and when there is a disconnect between what the patient wants or needs and the view of the family or relatives. In contrast, professionals often witness the difficulty for relatives to participate in making decisions for somebody else, especially when children are at risk. Participants also mentioned the impact that wearing special equipment during the pandemic had on the ability to communicate with patients and families [e.g., 47,56,69,74–76].

**Support network.**   The need for a circle of support was expressed in many ways. Support could come by own relatives and by peers. The impact of isolation was reported in studies covering the CoVID-19 pandemic when professionals had to focus on protection for self and others. Peers can be a second family, especially when working together for several years, and being separated from them because of re-location can be stressful. The feeling of isolation can also arise when the buffer provided by families and friends is missing, or they feel own relatives don't understand the challenges they go through daily [e.g., 38,45,46,72,85].

### Organisational causes and triggers

Some causes and triggers related to the actual or perceived hierarchy of the organisation. This category includes elements related to hierarchy and seniority, workload and roles, trust and support, regulations and procedures, culture of the context, and resources like equipment and staffing. Again, some of these triggers were more evident during the pandemic, for example, the introduction of new regulations and lack of resources.

**Hierarchy and seniority.**   Strict hierarchy and poor communication among different levels, or collaboration among people covering different roles is often a source of distress [e.g., 51,65,67,73]. This often relates to doctors and nurses, but also to other members of the organisation. For example, nurses describing their frustration at physicians not valuing the nurses' opinion, or discrepancies in approach and goals between different types of professional role (secondary care–to treat and discharge a patient- versus social care–to manage a patient at home long term). Sometimes roles bind senior professionals to follow protocol and regulations as dictated by policies that lower ranked staff may not comprehend. At individual level, this subcategory could overlap with personal risk when an autocratic leadership can be seen as bullying or harassing, and professionals feel victimised by the culture. Early engagement and good communication became crucial for the buy-in of regulations and procedures at all levels.

**Workload and roles.**   Participants described workload and the multitude of roles that staff must sometimes balance as a cause of stress [e.g., 50,65,69]. As key factors, workload and roles are connected; a professional may cover multiple roles (managing care and staff), and this add to their workload and burden. Or, as in the pandemic, circumstances require that staff members cover roles that differ from what they were trained for.

**Trust and support.**    Another cause of moral distress linked to organisational causes is the perception of being under-supported or a lack of trust. This was described in terms of feeling let down by the organisation and often a sense of deception or betrayal that may lead to trust issues. Feeling ignored when speaking out about decisions that went against their moral values, the lack of accountability of others, and the lack of support from management and senior staff were named as examples of what was felt as deception. Moreover, professionals can feel isolated when conflict with peers arises and displays tension due to different moral values [e.g., 46,51,56,61,78].

**Regulations and procedures.**    Regulations and organisational procedure can also be a cause of moral distress. Where the system impedes delivery of care in line with patients' wishes and values, this can make the work of HSCW more difficult or force them into making unfair, unjust or unethical decisions, as they perceive them. Recently, evidence of moral distress linked to wider system regulations, like policies, has been published, highlighting the impact of health inequalities on professionals' wellbeing when policies restrict the options HSCW can offer to their patients (inequalities of access to care or treatment) [e.g., 65,73,78,85].

**Culture of the context.**    Regarding employing organisations, many described issues around the environment of their workplace, the culture and resources which were (or were not) made available and a lack of staff, especially at busy times. The culture of a workplace, the attitudes and system principles, can have a great impact on staff feelings and behaviour [e.g., 41,50,59,83,84,86]. Participants mentioned a culture of blame, double checking on how well somebody did their work, overstepping into their role, and fear of repercussion or fear to speak up (psychological safety). These causes of distress can also lead to professionals deciding to change organisation or leave the profession.

**Resources: Equipment and staffing.**    Shortage of resources, like appropriate equipment or medications, and personal protection materials is described as a source of distress because it hinders the ability to care properly for patients whilst also protecting oneself. Shortages in human resourcing has an impact on individuals' workload and therefore is an indirect cause of distress [e.g., 38,42,45,71,78,82].

## Themes

Subcategories of causes and triggers of moral distress were aggregated to generate themes. Aggregate themes are shown in Table 4 and were identified as:

- *Be in someone else's shoes*–Empathy is one of the abilities care professionals have, to imagine what others may experience. Professionals looking after patients with poor quality of life or facing painful treatment procedures with no assurance of good results, may feel distress. This is related to their own moral values and thoughts of being unable to do it right. The welfare of others is their priority, and this sometime creates an inner conflict between what they know is necessary practice (due to processes and guidance on treatment) and the position they cover as patients' advocates.

- *Experience leads to autonomy and increasing responsibility*–HSCWs start their professional journey learning from training and daily practice. They witness senior team members making decision they may not agree with and feel frustration. With the years, they learn how decision making is linked to many elements of care: resources, procedures, and capabilities. Their own responsibilities and workload increase, and so the pressure and likelihood and magnitude of distress.

- *What comes first? Me, the patients, the organisation*–Beside the professionals' duty to care and their belief that patients come first, their experience leads to understand that

organisational priorities often dictate procedures and priorities. In circumstances like the CoVID-19 pandemic, professionals also faced risk to their own safety, making them question who should come first, and feeling that their abilities to cope with caregiver burden were stretched too far.

- *Counting on others*–HSCW often feel the need to be understood when processing their stress. Their immediate support network, families and friends, may not always understand what they go through due to lack of experience and knowledge. Peer-to-peer support becomes important as a coping mechanism but relies on sharing the same moral values. When the organisation fails to support their employees, because management ignores their needs or people are not accountable for their actions, professionals feel they are let down.

- *Top-down approach*–The culture of the workplace has a huge impact on employees. If people at senior level or in management don't address hostile behaviour but favour unhealthy competition, don't consult or engage with the wider team when making decision or don't address patients' needs properly but favour profit, professionals feel trapped and not valued.

- *Struggling to meet demand*–Staffing and resources, especially in emergency situations like influenza periods or pandemics, are crucial to assure professionals are not overloaded and can do their job properly. When workload stretches HSCWs abilities to meet demand, they feel they are letting people down.

## Analytical themes

Professionals have their own set of moral values that enter in conflict or concordance with the values of other people he or she interacts with. Health and social care professionals can experience tensions between what they think is right and what patients and relatives want. This tension triggers an internal conflict that could be resolved following regulations and being supported by other members of staff. When regulations fail to help decision making in accordance with moral values, or peer-to-peer support is lacking or, even worse, conflict arises, a person is likely to feel isolated and lost. Moreover, if the external circle of support is disconnected, because of pandemic circumstances for example, the HSCW is missing a buffer, an opportunity for copying with the distress.

In the health and social care environment, there is a hierarchy of leadership that impose regulations and culture, at micro level like in wards and departments, and at macro level, meaning the management of the whole organisation. The values that make the micro and macro levels function have an impact on the roles and duties of professionals. If those values are in contrast with staff members moral values, professionals feel a mismatch that can trigger emotional and physical responses of moral distress and injury. This causes professionals to question themselves and their sense of belonging to the organisation, this may have repercussions for their resilience and wellbeing, and employees' turnover.

## Secondary objectives

**Factors associated with moral distress and injury.**

**1. Psychological safety and lack of it as cause of moral distress and injury**

None of the papers mentioned this term specifically, however the texts were explored for content around the definition of psychology safety. The definition used for this purpose was as follows:

A shared belief that interpersonal risk taking is safe. A climate in which people are comfortable expressing and being themselves e.g., a work environment in which employees feel secure in speaking their minds without fear of retribution or embarrassment [90].

Using the definition above, implicit reference was made to psychological safety in terms of fear of the consequences of raising issues relating to ethics and morality. For example, in Ritchie et al. [75]: *"Participants expressed their desire to engage with management regarding ethical issues they encountered in their daily work. However, they feared that their input would be perceived as unnecessary or unwelcome, potentially exposing them to negative consequences."*

In addition, the term 'complicit' was often used when describing distressing situations where professionals felt they had to witness poor care and not speak up. For example, in Foster [50]: *"Rather than be vocal and say 'well it's not clinically indicated', and everything was going well, we have to play the game. And that's why you don't do it. Because you have to shut up and put your head down."*

Related to this, fear of retribution, not being heard and a sense of powerless in orchestrating change were frequently described, especially by those in more junior roles. This suggests that the working environment was not conducive to speaking out, certainly not universally. This may have the effect of exacerbating distress and increasing the likelihood of this continuing to injury [34].

### 2. Diversity/cultural differences and moral injury/distress

Articles including information on the influence of participants' characteristics and social determinants of health on moral distress and injury are included here. The hypothesis of diversity and culture as drivers of moral distress and injury was initially suggested by reviewers. However, two thirds of studies did not report participants' ethnicity and where it was mentioned, it was flagged as a potential cause of distress. Data on education and cultural elements (norms, symbols, language) were not collected. Only one paper [63] made a reference to religion as "remotely informing moral values" and a potential source of coping mechanisms rather than being a trigger or cause of moral distress.

Study sites were always indicated, North America and Europe, specifically England/UK were represented more than other regions. Studies were conducted across a number of different healthcare systems from the non-universal insurance system of the USA [40–42,44,48,57,62,63,69,76,78,85,87] to universal government-funded health system (single-payer) of Italy [43,64,65,67,86], UK [6,46,51,66,68,72,74,77,81], Sweden [45], Canada [47,49,52,54–56,61,70,71,73,75,79,82,84], Norway [58–60] and Australia [50], to the universal private health insurance system of Netherlands [39], and the universal public insurance system (social insurance) of Romania [53]. Although clear themes emerged that related to different healthcare systems and funding mechanisms for care (e.g., what is available to the individuals in that system based on resources and guidelines), overall, descriptions of events which could cause moral distress showed some consistency and are represented in the categories above.

### 3. Major events/disasters (e.g., pandemics, low probability/high impact events) and potential for influencing moral distress and injury experiences

Sixteen articles [38,46–48,51,57,62,63,65,66,71,77–79,82,85] specifically explored the role of CoVID-19 in HSCW and contribute to the identification of themes summarised in categories and sub-categories (Table 3). The pandemic appeared to impact the deployment of staff to various roles and clinical contexts. For example, nurses were sometimes given tasks or assigned to teams where they felt they were not sufficiently trained or competent [38,51]. In addition, several causes and triggers of moral injury and distress were specific to the CoVID-19 pandemic context. Unique management and administration-based conflicts emerged due to the new

arrangements and systems that were put in place [48]. Staff reported the need for management to act as a support network and be advocates for them:

*"Definitely, we need our managers to support us and be advocates for us and not just parrot what they think we want to hear."* Silverman et al. [78]

The pandemic also seemed to heighten issues around hierarchy and seniority, and the lack of control that lower ranking members of staff had over their work as in Sukhera et al. [82]:

"... pandemic policy was quickly developed within hierarchical power structures at a "higher level" (R02) ... getting "edicts coming down from much higher," as they were told "this is the new rule, and you have no input" (R03). ... in the context of a pandemic, "there's not really space for feedback" (R02)."

Another key element, relevant to CoVID-19, was in relation to regulations and procedures, and lack of evidence-based treatment, that were a consequence of the pandemic.

*"... nobody really knows what... the research is just coming out, so things just keep changing... parameters keep changing, first we're saying, okay early intubation, then we're like, no let's push off intubation, so all of that definitely affects us because you're trying to do the best, but things just keep changing."* Silverman et al. [78]

Policy related to personal protection equipment (PPE) was also a frequent topic of discussion in these studies. For some, PPE made delivery of care more challenging, in combination with other policies such as restricted visitation:

*"Policies and procedures were instituted to decrease transmission of the CoVID-19 virus, which included wearing of personal protective equipment and implementing a restricted family visitation policy. Both policies prevented the provision of high-quality holistic nursing care to patients."* Silverman et al. [78]

The other issue frequently discussed was the lack of PPE and staff safety:

*"And we weren't wearing masks. [. . .] And then as research has gone on and they said now you need all this PPE, well we didn't have that and it feels like they knew that we should have had it, but they just weren't gonna say it 'cause they didn't have the equipment and it didn't matter 'cause it was only us going in there."* French et al. [51]

Finally, participants reported that the need for lone-working and the effect of reducing team cohesion and support, led to an increase in isolation which exacerbated the effects of moral distress. For example, in Liberati et al. [66]:

"We have our morning meetings, but when we're in the office together we spend a lot of time discussing things, mulling things over, working things out with each other, and, yeah, we can pick up the phone, but it's not the same. It's not as readily available. So, the decisions are being made much more in isolation. So, there isn't that check and balance."

**Participant-reported coping and treatment strategies and/or interventions.** Thirty Nine studies [38,41–49,51,54–60,63–71,73–78,82–84,86–88] explored coping strategies and

Table 5. Areas for preventing and addressing moral distress and injury in professionals.

| Personal Solutions and Coping Mechanisms | Social Solutions | Organisational Changes and Interventions |
|---|---|---|
| Health and wellbeing | Nurturing relations with family and relatives | Education |
| Reflection | Friendship | Resources and infrastructure |
| Separating life and work | Peers support | Culture and communication |

positive action for the purposes of preventing or treating moral distress or injury. Both adaptive and maladaptive coping strategies were discussed, as well as effective and ineffective solutions.

A comprehensive list of coping strategies can be found in Hancock et al. [54]. What unites HSCWs against moral distress and injury is the search for a quick and easy fix that can make the workload bearable in the short term (e.g., [78]). Some participants managed on their own, some preferred social interactions and peer support (e.g., [65,69]).

The quotations of all thirty-nine articles were used to provide insights on strategies. Solutions were discussed in terms of personal solutions and coping, social solutions, and organisational changes and interventions, as shown in Table 5.

**Personal coping strategies:** Helmers et al. suggests a division between action and reflection when discussing coping strategies. Seeking more information regarding the patient's case and course of action was described as an action. Similarity, choosing to address the distress with time off work or self-care. Reflection was described as re-focusing one's perspective [55]. For example, some participants described their coping by means of numbing themselves to it, distancing themselves from the situation and moving on. They described this as necessary for continuing to do their job. Some participants described their distressing experiences as just part of the job. In Robinson and Stinson [76]:

"We just ignore it. I have another patient to care for. And we just move on."

However, for many, with repeated exposure and increasing experience, came personal growth. In French et al., participants spoke about new perspectives, a sense of greater influence and an ability to cope with their role and experiences [51].

"A significant element of growth has happened from the experience I would say. And a feeling that actually yeah, I do have some power in these situations [. . .] there are situations that I can have some influence over, you know in line with my values."

Some studies detailed how participants needed to draw a distinction between work and home life as a way to move on and be able to live their lives, as described in Silverman et al. [78]:

"I have a fairly good ability to, kind of, compartmentalize and leave work at work, and I can talk with my dad about certain things because he understands the constant stress, he understands what it means, so I can get a little debrief."

The CoVID-19 pandemic situation put an emphasis of work and life balance for professionals. In Howard [57], a professional suggests:

"The biggest thing that we should do. . . is to remember that there still is a work-life balance, even if you're in the middle of a worldwide pandemic. You can't pour from an empty cup."

Faith and spirituality, and humour, were also described as ways of managing distressing situations, as in Matthews & Williamson [68]:

"I think the only way to cope with the stress of the job is to have a laugh with your colleagues and hopefully the patients when they are in good moods. If you didn't laugh I think you would either cry or go mental yourself."

**Peer (social) and organisational support** networks were described as helpful, especially within teams, to de-brief and ensure understanding and shared awareness of the experience of caring; a useful forum to discuss and share thoughts.

In Joolaee [61], the importance of the understanding that comes from peer support is expressed as

"Having people understand what I have been going through–helps."

In Scott [77], a nurse explains the need for familiar relations in difficult circumstances:

"They did a supported de-brief and there were tears because of the deaths, it was really emotional. I said to the staff I need to go and find 'my team' if that's okay. I need to go and find them."

Investing resources for ethical training to establish an ethical climate as well as adequate training in morally distressing situations were also suggested to "*combine moral sensitivity and ethical climate to enhance moral agency*" [43,78]. Changes to regulations and organisational adjustments were suggested by participants such as: effective communication within teams [74], ways of increasing time between patients to allow for "breathing room" [42], integrating family meetings [69].

## Discussion

Conversations around moral injury have become more widespread since the start of the CoVID-19 pandemic, a number of qualitative studies giving accounts of health and social care workers experiences of moral distress have been published. Prior to this, the experience of moral distress and injury was explored across a number of different settings, especially the military service, originating from the philosophical works of Jameton [2]. He understood moral distress in HSCW as a two-stages process manifesting as guilt for lack of agency to make the morally acceptable action and the distress of not being able to act on those feelings. Therefore, moral distress relates to the moral values of professionals who deal with situations that challenge their moral self, that is the morality of who a person is and how a person acts [91]. An explanation on the building blocks of the moral self is beyond this review and can be found in Jennings et al. [91]. Results of this review of fifty-one articles indicate that the experience of moral distress and injury is familiar to HSCW, and they are often aware of how they are affected by it. HSCW are faced with daily choices and actions that involve their moral reasoning, that is *reasoning based on evaluative judgments pertaining to others' welfare, rights, fairness, or justice* (p3. [89]). When professionals' actions and choices relating to clinical care are in opposition to their moral judgement, internal conflict occurs.

The aim of this review was to explore the lived experiences of moral distress and injury among HSCW to identify reasons and triggers. Fifty-one articles were selected which describe the potential causes and triggers of moral distress. Only five focused specifically on moral injury but presented similar elements when reporting potential reasons for it [46,51,64,66,71].

The causes and triggers of moral distress and injury named by professionals were divided in three main categories, according to their link to the individual, social, or organisational domain. Patients' care (individual), HSCW's advocacy role (social), and the culture of the context (organisational) were the most referenced causes of moral distress, followed by morals, beliefs and standards, and regulations and procedures. Clear boundaries between categories were not always definitive. Elements like lack of time, poor planning, and ineffective communication, were difficult to assign to a particular domain, because they can be attributed to personal abilities and to organisational practices. The themes arising from the direct quotes from participants are linked to their empathy, experience, priorities, need for support, and the context their work in, common sources of organisational stress [92]. This is indicative of the influence organisational structures and values can have on individuals. The moral values underpinning organisational choices are filtered through a hierarchy of individuals who place their own moral judgment upon processes and approaches and then cascade to lower ranking professionals. Depending on roles, length of experience, and team dynamics, health and social care workers can feel in distress if they are prevented in taking, or expected to work against, what they think is the right course of action. Moreover, professionals are people with their own morality, and they interact with patients, families and colleagues within the health and social care work environment, influencing the moral principles and culture of the context. Therefore, internal and/or external conflict may arise if professionals' values are not aligned with organisational ones. This can happen when financial constraints, poor planning, or top-down decision making on patients' care influence daily practice. Professionals will experience moral distress, especially if social and peer-to-peer support is lacking, or their own safety is at risk. In an organisational structure where individuals often develop their own strategies to cope with daily emotional conflicts, there needs to be a culture of support and employees need to feel valued.

A shift between early articles (before CoVID-19) and those published more recently (after the pandemic) (Table 2) was noticed in assigning the responsibility of moral judgement onto the individual versus the need for the organisation to invest on an ethical and supportive culture. Participants in earlier articles refer to the burden of making decision on their own, based on their own knowledge and experience, finding in themselves the ability to cope with that responsibility. In later literature, moral distress is perceived as an organisational issue, where priorities are dictated, with the expectation that staff should accept them. The shift is also recognised in the switch from discussing adaptive and coping mechanisms, where the onus is on the individual to cope, to the need to tackle the symptoms and mitigate the impact collaboratively. The results of this review highlight the need for patients' care to be a team affair. Professionals should be aware of the reasoning behind treatment decisions and should be able to express their concerns, receive training and guidance during decision making, whilst also being offered support. Families should be kept informed and engaged when a shared decision has been made with the aim of providing a consistent message. The necessity of rising awareness of moral distress and injury, the need for trust and support among colleagues, and an understanding of the visible signs of moral injury, as well as a safe space to vent concerns, have also emerged as key themes (Table 3). Senior staff should be trained in identifying moral injury pre-emptively and introduce the use of validate scales to prevent and plan educational interventions [48,93].

The secondary objectives of this review, to understand the impact of specific factors, like psychological safety, diversity and special circumstances, were partially explored in the studies selected. Implicit reference was made to the important of feeling safe to speak up [50,75]. Religion was investigated by only one group of researchers and was named as coping strategy rather than affecting the experience of the participants [63]. As anticipated, the special

circumstances that occurred during the CoVID-19 pandemic were often referred to as causes and triggers of moral distress. The findings of this review are consistent with reports in the media and academic research, highlighting the effects that understaffing, scarce personal protection measures, changing working environment, and evolving regulations had on professionals' mental health during CoVID-19 [1,94]. The impact of lack of resources was a reoccurring theme for the participants reporting their experience of the pandemic. HSCW were aware of the moral distress they were experiencing but they didn't feel free, or have the time and space, to tackle it [43]. To cope, HSCW had to emotionally distance themselves from decisions and actions that did not align with their personal views and reported "feeling empty" because of a repeated inner trauma [58]. Linked to this, was the fear of retribution for speaking out in opposition to leadership decisions, which led to lower ranked staff conforming to practices they may not agree with [45]. In contrast, there was evidence that moral distress can increase resilience, which was mentioned in several papers (e.g. [54,55,64]), and the pandemic showed how resilient HSCW can be [95].

This synthesis may serve to inform professionals and managers about strategies and mechanisms for the mitigation of possible consequences on staff wellbeing and retention. The coping mechanisms described by HSCW detail the variety of adaptive and maladaptive approaches that they engage in, to continue in their role. In contrast, when coping leads to a sense of empowerment and motivation, this acts as a buffer to future harm. From action to reflective strategies and avoidance approaches, HSCW and managers are trying hard to find solutions at individual and organisational levels. This means they acknowledge moral distress and injury do occur more often than assumed before the pandemic [9], and that experience was exacerbated by the high demand and low resources during the CoVID-19 pandemic, coming to the surface for institutional acknowledgement. Several organisational level strategies to identify undue distress and mitigate against moral distress and injury have been documented. A need for such strategies was made even more evident during the pandemic (see implications for practice). If those strategies were implemented widely, there is some evidence the negative impact of moral distress on staff wellbeing might be reduced with positive consequences on quality of care [96].

## Implications for practice

Moral distress and injury can be caused by a combination of factors at the individual, social and organisational level, and all should be considered when looking at approaches to dealing with it. To address the individual and social aspects, it is important that healthcare workers have an awareness of moral distress and can identify this in themselves and colleagues, preferably at an earlier stage and at a point when moral injury is better prevented. Risk factors for developing moral injury have been identified in the causes and triggers of moral distress and this knowledge could be used in creating educational materials for use within HSCW education and ongoing training to help them cope with and address it. This could incorporate how to report moral distress and injury at an institutional level and how the organisation and colleagues can support individuals, building a supportive network. At an organisational level, managers should focus on ensuring that clear information about regulations and procedures, especially when these have changed, are promptly circulated to staff, including consultation sessions to listen to staff concerns where possible and appropriate. Moral approaches to leadership, which encompass ethical, authentic and servant leadership styles, have been shown to positively impact organisational outcomes [17]. Creating an environment where professionals are safe to speak up, where they feel empowered and can achieve an optimal work-life balance could prevent mental health issues arising during challenging situations. For example, a

"psychological first aid" support team, to be available when staff require support, has been advocated by professionals alongside better access to resources like ethics consultations and chaplaincy [42]. Those reporting distress (or identified by colleagues as suffering), should be offered specialist support, time and space to work on ensuring this does not escalate further, as suggested by a Welsh Government report on moral injury in healthcare workers [97]. This report also confirmed the utility of active monitoring programmes, due to issues with health-care workers not always seeking support.

## Limitations of the included studies

This review, unlike others looking at moral distress and moral injury, focused on published qualitative studies, capturing the narrative which is often missed in quantitative exploration. Systematic reviews are, of course, dependent on the quality and nature of the evidence available. The evidence reported in this review varied greatly in terms of quality of reporting, although overall the body of work was of good or better quality. A significant limitation of many studies was the lack of accurate reporting of sample characteristics such as ethnicity, age, gender and in some cases even the sample size. This means that there can be limited discussion of how the causes and triggers of moral distress varies according to these characteristics. It is likely that the experiences of those HSCW from differing ethnicities and genders would be different, as demonstrated with the mental health impact of the pandemic on HSCW [98–102].

The method of collecting data, understandings and reporting of theoretical approaches and analysis techniques were also varied and sometimes inconsistent with recommendations. Many studies also reported issues presented by COVID-19 which meant that in person interviews (and the personal nature that was preferred for interviews) were no longer possible, meaning that some information could potentially be withheld or overlooked. Many researchers recruited opportunistically, or participants were self-selected which is likely to introduce bias. Other identified sources of bias were recorded: lack of reported reflexivity in recruitment and collecting data, social desirability (if respondents thought their answers could affect appraisals), and the rapid and changing nature of the pandemic meaning that experiences might have evolved.

## Limitations of the review

By limiting this review to published and accessible literature, research in Western countries and papers published in English and Italian, some papers may have been missed. Expert elicitation during searches was not employed, however due to the depth and breadth of the research available, selection had to be made for practical reasons. Seven people, with different backgrounds, worked on article selection, coding and analysis of quotations. The process was iterative, and searches were repeated, and additional papers found. The different levels of experience in conducting literature reviews and synthesis, whilst working collaboratively through the process might have affected consistency of selection and coding, but the breadth of backgrounds and expertise enriched the results of the review. Moreover, the conclusions drawn from this review are strengthened by the inclusion of a large number of studies, participants, viewpoints and designs. To avoid reiterating the views of the publishing authors, reviewers have re-analysed and interpreted the data based on participants own words and direct quotations.

## Implications for future research

Future research and reviews should focus on exploring the role of culture, ethnicity, religion, and other social determinants of health. Influence of different health and social care systems

on moral distress and injury should also be examined in to order to address any potential causes of distress for professionals. In addition, studies that evaluate the impact of roles and seniority would be helpful to understand how professionals with differing backgrounds experience morally distressing situations. Contextual factors like organisational values and priorities, practices that mitigate the development of moral distress like the involvement of ethics committee or inter-professional support, could be explored with methods such ethnography, participatory action research or realist evaluations. Indeed, there is a clear need to further evaluate strategies and interventions to help reduce distress when it occurs, notice it's emergence and reduce its impact on workforce retention which emergent situations can exacerbate. Interventions from other settings have included techniques that focus on self-conscious emotions, such as shame and pride [11,103], self-compassion, acceptance and resilience [94,103] and psychotherapeutic interventions such as adaptive disclosure [104]. Such methods should be tested in the health and social care setting. In addition, full disclosure of participants characteristics as ethnicities, ages, gender, roles, educational levels and experiences, especially when they may influence the results, will enable further analysis and improve reviews' quality and impact. More research is required to examine the effects of organisational culture and professionals' diversity on the risk of developing moral distress and moral injury to tailor educational material and increase awareness and support.

## Conclusion

Moral distress and moral injury in health and social care professionals are associated with individual, social and organisational factors. Professionals lament that time constrains that hinder patients' care, economical choices that reduce resources regardless of needs, and organisational priorities that don't reflect individuals' values affect their mental health. The unique effect of the CoVID-19 pandemic exacerbated the distress of a workforce already experiencing some personal conflict. This confirms that the increased pressure on individuals due to lack of resources and top-down organisational changes push staffs' ability to cope to the limit. An environment where professionals would feel psychologically safe to speak up and supported may help to mitigate moral distress symptoms and prevent moral injury.

## Supporting information

**S1 File. PRISMA checklist.**
(DOCX)

**S2 File. Protocol.**
(DOCX)

**S3 File. Table of included studies.**
(DOCX)

## Author Contributions

**Conceptualization:** Emily S. Beadle, Agnieszka Walecka, Annalisa Casarin.

**Data curation:** Emily S. Beadle, Agnieszka Walecka, Amy V. Sangam, Jessica Moorhouse, Matthew Winter, Annalisa Casarin.

**Formal analysis:** Emily S. Beadle, Agnieszka Walecka, Amy V. Sangam, Jessica Moorhouse, Matthew Winter, Helen Munro Wild, Daksha Trivedi, Annalisa Casarin.

**Funding acquisition:** Emily S. Beadle, Annalisa Casarin.

**Investigation:** Emily S. Beadle, Amy V. Sangam, Jessica Moorhouse, Matthew Winter, Helen Munro Wild, Daksha Trivedi, Annalisa Casarin.

**Methodology:** Emily S. Beadle, Agnieszka Walecka, Amy V. Sangam, Jessica Moorhouse, Matthew Winter, Helen Munro Wild, Daksha Trivedi, Annalisa Casarin.

**Project administration:** Emily S. Beadle, Annalisa Casarin.

**Software:** Helen Munro Wild, Daksha Trivedi.

**Supervision:** Emily S. Beadle, Agnieszka Walecka, Daksha Trivedi, Annalisa Casarin.

**Validation:** Daksha Trivedi.

**Visualization:** Emily S. Beadle, Matthew Winter, Helen Munro Wild, Annalisa Casarin.

**Writing – original draft:** Emily S. Beadle, Agnieszka Walecka, Amy V. Sangam, Matthew Winter, Helen Munro Wild, Annalisa Casarin.

**Writing – review & editing:** Emily S. Beadle, Agnieszka Walecka, Amy V. Sangam, Jessica Moorhouse, Matthew Winter, Helen Munro Wild, Daksha Trivedi, Annalisa Casarin.

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
