## [Decision Letter · Decision Letter 0]

10 Dec 2023

PONE-D-23-19884Experiences of moral injury in healthcare workers: A rapid systematic review of qualitative studiesPLOS ONE

Dear Dr. Beadle,

Thank you for submitting your manuscript to PLOS ONE. After careful consideration, we feel that it has merit but does not fully meet PLOS ONE’s publication criteria as it currently stands. Therefore, we invite you to submit a revised version of the manuscript that addresses the points raised during the review process.

We look forward to receiving your revised manuscript.

Kind regards,

Nancy Clark, PhD

Academic Editor

PLOS ONE

Journal Requirements:

2. We noted in your submission details that a portion of your manuscript may have been presented or published elsewhere. "This work is a review, as such the data is compiled from the reviewed papers and quotes are published in these respective papers, cited appropriately where used." Please clarify whether this [conference proceeding or publication] was peer-reviewed and formally published. If this work was previously peer-reviewed and published, in the cover letter please provide the reason that this work does not constitute dual publication and should be included in the current manuscript.

3. Please upload a new copy of Figures 1 and 2 as the detail is not clear. Please follow the link for more information: " ext-link-type="uri" xlink:type="simple">https://blogs.plos.org/plos/2019/06/looking-good-tips-for-creating-your-plos-figures-graphics/"
https://blogs.plos.org/plos/2019/06/looking-good-tips-for-creating-your-plos-figures-graphics/

Additional Editor Comments:

We have received to independent reviews and the decision has been made for major revisions to your article. We look forward to your response.

Thank you,

Nancy

Reviewers' comments:

Reviewer's Responses to Questions

**Comments to the Author**

1. Is the manuscript technically sound, and do the data support the conclusions?

Reviewer #1: Yes

Reviewer #2: Yes

2. Has the statistical analysis been performed appropriately and rigorously? 

Reviewer #1: Yes

Reviewer #2: N/A

3. Have the authors made all data underlying the findings in their manuscript fully available?

Reviewer #1: Yes

Reviewer #2: Yes

4. Is the manuscript presented in an intelligible fashion and written in standard English?

Reviewer #1: Yes

Reviewer #2: Yes

5. Review Comments to the Author

Reviewer #1: In general, it is important to provide an overview of the experience of moral distress and moral injury, causes, and the consequences among health care workers especially with the overview of qualitative studies, insights into the experience of health care workers can provide a deeper understanding of the constructs of MI and MD. Nevertheless, the manuscript needs to be fundamentally revised.

For details see attachment

Reviewer #2: Thank you for allowing me to read this manuscript, PONE-D-23-19884. Experiences of moral injury in healthcare workers: A rapid systematic review of qualitative studies

1 1. The study presents the results of the original research.

Yes, the study is a rapid review of qualitative studies.

As a reader, the introduction/background leads me to the purpose. This rapid systematic literature study examined Experiences of moral injury in healthcare workers. A literature review's strength is adding knowledge to the current database of knowledge.

2. Results reported have not been published elsewhere.

Not the result of this review.

3. Experiments, statistics, and other analyses are performed to a high technical standard and are described in sufficient detail.

The method section is well described and follows recommended steps. The Prospero should be added to the list of references, as well as the CASP.

Figures, tables and Flowchart are precise. Quality assessment, according to CASP, is clear.

The analysis: Please clarify what kind of analysis you have performed. Thematic analysis, according to Thomas and Harden? The references state the synthesis should be performed via three stages which overlap to some degree: the free line-by-line coding of the findings of primary studies, the organisation of these 'free codes' into related areas to construct 'descriptive' themes, and the development of 'analytical' themes. Another reference you refer to is Sandelowski and Leeman, and they state consequences, etc, are, at this point, simply words used to encompass segments of data researchers saw (i.e., coded) as belonging together. Although a single word such as consequence "may name an idea, [that word] does not operate as an idea until it is put into a sentence or assertion. An idea/theme needs a subject and a predicate before you can use it as a basis of understanding."

Result: Figure 2 needs to be more readable, with too much text and too small figures.

Definition of moral distress and moral injury should be placed in the introduction- that will guide analysis later on.

Presenting the participant's view?? Does this connect to the primary aim?

Interesting findings/themes. However, are they themes? Causes and Triggers, Experiences. Consequences. According to the reference you use—see above. Are they not simplified to answer the aim of identifying reasons for and factors that influence moral injury and distress in HCWs? According to Thomas and Harden, you are supposed to go below the text to be able to describe the theme. Reading the quotations, there is more than one word labelling the theme, saying nothing.

Cause and triggers are also about experiences.

The second aim moves forward better, but the themes are left out, and the research questions are used as theme labels.

What kind of analysis and synthesis is this?

The analytical themes are good. So please connect them with the descriptive themes. Now, there is a gulf between them.

Discussion. There is much repetition of the result- referring to the included studies.

I am not surprised that your result agrees with other studies and media. So, what new understanding or knowledge did you bring forth?

4. Conclusions are presented in an appropriate fashion and are supported by the data.

The author states that this review has highlighted how healthcare workers experience moral distress and injury in their own words. Details of the roles of individual, social and organisational factors in causing distress and moral injury, the experience and the associated emotions, the consequences of experiencing moral conflict and how HCWs cope have been reported. This is just another way to present the objectives. The research questions used were entirely quantitative, and this conclusion suggests that this review has highlighted that more research is required to examine the effects of culture and diversity on the experience of moral injury and distress. (once more a quantitative perspective). The conclusion should have highlighted the analytical themes.

5 The article is presented in an intelligible fashion and is written in standard English.

Yes

6. The research meets all applicable standards for the ethics of experimentation and research integrity.

Yes

7. The article adheres to appropriate reporting guidelines and community standards for data availability.

Style and format OK Clarity -Yes

8. Other comments: Advice for improvement of the manuscript?

There is a need to clarify the method section and present the result.(then discussion etc will also be improved)

There are 92 references used in this paper, and out of these, 20 are old, ten years or more; 19 are used in the introduction and out of these, 6 are about 10 years old. I cannot find the year of publication in the list for references 5,78 and 87.

This manuscript would contribute to our knowledge base if the data were analysed and handled according to the references used. Ia m looking forward to see an amended manuscript

6. PLOS authors have the option to publish the peer review history of their article (what does this mean?). If published, this will include your full peer review and any attached files.

Reviewer #1: No

Reviewer #2: No

---

## [Author Response · Author response to Decision Letter 0]

28 Feb 2024

Reviewer’s comments to the authors:

Editor comments:

We strived to comply with this, thanks

2. We noted in your submission details that a portion of your manuscript may have been presented or published elsewhere. "This work is a review, as such the data is compiled from the reviewed papers and quotes are published in these respective papers, cited appropriately where used." Please clarify whether this [conference proceeding or publication] was peer-reviewed and formally published. If this work was previously peer-reviewed and published, in the cover letter please provide the reason that this work does not constitute dual publication and should be included in the current manuscript.

We thank the editor for this comment, and we want to clarify that, this being a review article, the content of the manuscript (beyond the quotes published in relevant papers) is not published anywhere else in any format. 

3. Please upload a new copy of Figures 1 and 2 as the detail is not clear.

The figures were reformatted, and we changed figure 2 and added figure 3 as a result of the work on the content based on reviewers’ comments. The figures are now formatted as required. 

Reviewer 1:

In general, it is important to provide an overview of the experience of moral distress and moral injury, causes, and the consequences among health care workers especially with the overview of qualitative studies, insights into the experience of health care workers can provide a deeper understanding of the constructs of MI and MD. Nevertheless, the manuscript needs to be fundamentally revised. For details see attachment

We thank the reviewer for the thorough review of the paper and for the comment on the importance of the topic and its relevance in current times when accounts of professionals’ distress are many. We agree the revision of the manuscript was necessary to improve the impact it will have on the body of literature on moral distress. We revised it thoroughly and made further improvements, triggered by the reviewers’ comments.

Title

Only MI is addressed in the title. Why is MD not mentioned here? Perhaps it would be better to mention MD as well, since the main studies also use only this terminology?

The title has been revised and now mentions the modified focus. Moral distress is now added. 

Introduction

Line 78: Please add a short definition of Compassion Fatigue. We thank the reviewer for this comment, this was addressed, and definition included.

In the definition of MI the references are completely missing (Line 80-86). Which definitions were used here? We thank the reviewer for this comment, this was addressed, and references included.

Aims and objectives

The secondary objectives differ from those available online in the review protocol. Please adjust the information. We thank the reviewer for this comment, this was addressed, and a new protocol on Prospero has been created. It is currently under review as of submission, and a new link will be provided prior to publication. 

Method

Why was a Rapid Review conducted when the survey period was June 2021 to February 23? We thank the reviewer for this comment. As mentioned, we changed this from rapid to a systematic review and the methods section changed accordingly.

inclusion exclusion

Again, there is a discrepancy between the manuscript and the review protocol online. Please standardize here accordingly. We thank the reviewer for this and the comments below. This was addressed, and new details were added in the new revised protocol.

Were social workers included or excluded? Social workers were included, and this is mentioned in the text.

What are "health facility managers"? Why were they included and management not? We agree that this was misleading, and we changed the eligibility criteria accordingly.

Risk for bias/ quality assessment

The link for the CASP tool is not up to date. Please add a new link. We updated the link as suggested.

Results

Overall it would be nice if the presentation of results is something more precise and stringent. A brief summary of how many and which studies addressed each of the topics listed below might help. For example, in line 303 you wrote "few". Out of 46, only 3 mentioned MI. This is a result worth mentioning, (which may also have relevance to both the title of the review and the discussion at the end). Also, I would appreciate if individual factors of the dimensions and categories were discussed, as they cannot be exclusively assigned to one category (see below for a detailed note/description).

We thank the reviewer for these suggestions. Authors met and discussed how this manuscript could be improved in terms of narrative and content. We realised the most important addition to the body of literature our review is introducing is the thorough investigation of cause and triggers of moral distress. We re-arranged the results section and, as suggested, we discussed the categories and sub-categories. The issue of assigning elements to one category was also discussed and reported in the manuscript. A table with reference to how many files mentioned the sub-categories was added.

Theme 1: Causes and Triggers.

A lack of time is classified as a personal trigger. Why is this not classified as an organizational trigger? 

The issue of assigning elements to one category is real and we had conversations around those elements we could not assign easily. We changed the themes tree to a figure that indicates the overlapping of sub-categories. [Figure 3]

Line 402: Here, the lack of trust is classified as a trigger. In the literature, the experience of betrayal is the cause of the loss of trust (as an experience). Here I would just like to see a separation betrayal as cause for the loss of trust as experience or consequence.

We consider the element of trust and its link with betrayal. We hope the narrative now reflects this link.

Experiences

Lines 426/427 here it says that people describe their experience of MD and MI in metaphors. Can't that storytelling also be understood in the meaning making context and thus as coping or learning? Perhaps one could at least discuss the point.

Sometimes the quotes are also not meaningful line 434 " plenty of emotional distress" does not characterize the distress in detail. Perhaps a statement could be picked out here which really characterizes emotional experience.

If the specific negative emotions were mentioned in only a few of the studies examined, this would also be an interesting result, but this result should be named clearly. Maybe as “non-specific negative”?

In the quote from line 462/ 463, questions are raised not only about the moral " role" but also about the moral self. This could then be mentioned in the text/graphic as well!

We decided to change the focus of the review and report the cause and triggers of moral distress/injury. We noticed, from reviewers' comments, that we had too much content and we could not address properly the numerous concepts that ‘’experience’’ and ‘’consequences’’ constructs entail. We decided to restrict the report and look to those elements that did not have the same amount of literature published on. The moral self was also mentioned and discussed briefly.

Consequences

Please add references (Lines 491,499)

Line 511: Here short-term experience and longer-term experience are mixed up. Please have a look at the syndrome perspective of Jinkerson or the conceptualization of Litz et al.

Learning from previous experiences can also be coping. Probably discuss this issue (lines 522-524)

Similarly to the above comment, we refocused the review report to address a gap in literature about a review of potential causes and triggers.

Secondary objectives

Diversity/cultural differences and moral injury/distress.

Here they only looked at triggers for MD and MI. Have differences been found in the experience of MD and MI? Are there differences in emotions or differences in physiological response? 

We replied to this query adding a comment in the results section that assumptions were not possible since we could not retrieve data linking participants characteristics and MD/I.

Major events/disasters (e.g., pandemics, low probability/high impact events) and 574 potential for influencing moral distress and injury experiences.

Line 581 please add a reference 

Thanks for this suggestion, we added a reference.

reported coping and treatment strategies and/or interventions.

Please add a reference at line 628. Also, from 666-667 it would be interesting to see the articles that have examined this. 

Thanks for this suggestion, we added references as needed.

At 653 it is said that ethical training can be useful or not. How do the authors see this, how do they justify this and please also a reference. 

The coping strategies section has been reworked according to the three main categories of individual, social, and organisational domain to reflect the causes and triggers section. Therefore, some concepts or conflicting messages are not present anymore. 

Discussion

In this section, it would be desirable to discuss the conceptual difficulties of experiencing, consequences, and coping strategy. These become especially evident in the emotional experience, in the social consequences (withdrawal or job abandonment), or in the reinterpretation (once in the experience and once in the learning). It would be good if at least the difficulty of the exclusivity of the categories could be addressed.

In line 694 I would like to have a reference source again. 

Line 727 here talks about the buffering effect of resilience. Resilience also includes other components that I think could be addressed here.

This section has now been reworked according to the analysis reported in the results section.

 Implications for practice

Almost the entire section lacks references to the implications that have already been made in the existing studies. Please add them.

We decided to go beyond the findings and suggestions of reviewed papers and focus on participants’ quotes. The new analysis enabled us to reframe the causes and triggers, and go deeper in finding aggregate themes. We framed the implications for practice on those themes.

Formal aspects: 

These comments were considered; we were not able to retrieve the lines of the quotes from original papers.

Please check the whole manuscript for spelling (753).

Please use consistent spelling (e.g. lines 41,95; 115,120,575,682). Please write out the constructs at the first mention and the abbreviation in brackets (e.g. line 90,170).

It would be nice if you could unify the headings in the style (line 233, line 243) (348 or 421).

For the citations in the results section, it would be nice if you also refer the lines of the quotes.

Formal aspects of tables and graphics 

These comments have been taken into consideration; it is not possible to know how the table will be split into the final format, therefore we will consider this when editing personnel will inform us about the issue of headings.

Please add a legend to table 2. This could explain the terms ICU and IPA, MD and MI. It would also be nice if the headings of the columns were repeated on each page to improve readability.

Reviewer 2:

Thank you for allowing me to read this manuscript, PONE-D-23-19884. Experiences of moral injury in healthcare workers: A rapid systematic review of qualitative studies 

1. The study presents the results of the original research. 

Yes, the study is a rapid review of qualitative studies. 

As a reader, the introduction/background leads me to the purpose. This rapid systematic literature study examined Experiences of moral injury in healthcare workers. A literature review's strength is adding knowledge to the current database of knowledge.

We thank the reviewer for these comments.

2. Results reported have not been published elsewhere. 

Not the result of this review.

3. Experiments, statistics, and other analyses are performed to a high technical standard and are described insufficient detail. 

The method section is well described and follows recommended steps. The Prospero should be added to the list of references, as well as the CASP.

Figures, tables and Flowchart are precise. Quality assessment, according to CASP, is clear. 

We thank the reviewer for this comment. We added the Prospero and CASP links to the references list. 

The analysis: Please clarify what kind of analysis you have performed. Thematic analysis, according to Thomas and Harden? The references state the synthesis should be performed via three stages which overlap to some degree: the free line-by-line coding of the findings of primary studies, the organisation of these 'free codes' into related areas to construct 'descriptive' themes, and the development of 'analytical' themes. Another reference you refer to is Sadlowski and Leeman, and they state consequences, etc, are, at this point, simply words used to encompass segments of data researchers saw (i.e., coded) as belonging together. Although a single word such as consequence “may name an idea, [that word] does not operate as an idea until it is put into a sentence or assertion. An idea/theme needs a subject and a predicate before you can use it as a basis of understanding. “

We thank the reviewer for this comment. It helped us revisit the section of methods and we hope it is now clearer.

Result: Figure 2 needs to be more readable, with too much text and too small figures. Definition of moral distress and moral injury should be placed in the introduction- that will guide analysis later on. Presenting the participant's view?? Does this connect to the primary aim?

We thank the reviewer for these comments. We now changed figure 2 (the alternative one is now figure 3) and erased the sections on definitions because they were not linked to the primary aim.

Interesting findings/themes. However, are they themes? Causes and Triggers, Experiences. Consequences. According to the reference you use—see above. Are they not simplified to answer the aim of identifying reasons for and factors that influence moral injury and distress in HCWs? According to Thomas and Harden, you are supposed to go below the text to be able to describe the theme. Reading the quotations, there is more than one word labelling the theme, saying nothing.

Cause and triggers are also about experiences. The second aim moves forward better, but the themes are left out, and the research questions are used as theme labels. 

What kind of analysis and synthesis is this?

The analytical themes are good. So please connect them with the descriptive themes. Now, there is a gulf between them.

Discussion. There is much repetition of the result- referring to the included studies. I am not surprised that your result agrees with other studies and media. So, what new understanding or knowledge did you bring forth?

We thank the reviewer for these comments. We reviewed the sections named above and focus on the revised primary aim. We made the analysis process and themes clearer, and defined the sub-categories the themes derived from. We separated the quotes for causes and triggers, adding a table, to make the reading easier and clearer.

4. Conclusions are presented in an appropriate fashion and are supported by the data.

The author states that this review has highlighted how healthcare workers experience moral distress and injury in their own words. Details of the roles of individual, social and organisational factors in causing distress and moral injury, the experience and the associated emotions, the consequences of experiencing moral conflict and how HCWs cope have been reported. This is just another way to present the objectives. The research questions used were entirely quantitative, and this conclusion suggests that this review has highlighted that more research is required to examine the effects of culture and diversity on the experience of moral injury and distress. (once more a quantitative perspective). The conclusion should have highlighted the analytical themes.

We thank the reviewer for these comments. 

We appreciated the

---

## [Decision Letter · Decision Letter 1]

24 Mar 2024

PONE-D-23-19884R1Triggers and factors associated with moral distress and moral injury in health and social care workers: a systematic review of qualitative studiesPLOS ONE

Dear Dr. Beadle,

Thank you for submitting your manuscript to PLOS ONE. After careful consideration, we feel that it has merit but does not fully meet PLOS ONE’s publication criteria as it currently stands. Therefore, we invite you to submit a revised version of the manuscript that addresses the points raised during the review process.

Thank you for the major revisions to the origianl manuscript. The manuscript has been significantly revised. Minor revisions are recommend to address reviewer 2 comments to the following sections provided below.  

We look forward to receiving your revised manuscript.

Kind regards,

Nancy Clark, PhD

Academic Editor

PLOS ONE

Journal Requirements:

Reviewers' comments:

Reviewer's Responses to Questions

**Comments to the Author**

1. If the authors have adequately addressed your comments raised in a previous round of review and you feel that this manuscript is now acceptable for publication, you may indicate that here to bypass the “Comments to the Author” section, enter your conflict of interest statement in the “Confidential to Editor” section, and submit your "Accept" recommendation.

Reviewer #2: All comments have been addressed

2. Is the manuscript technically sound, and do the data support the conclusions?

Reviewer #2: Yes

3. Has the statistical analysis been performed appropriately and rigorously? 

Reviewer #2: N/A

4. Have the authors made all data underlying the findings in their manuscript fully available?

Reviewer #2: No

5. Is the manuscript presented in an intelligible fashion and written in standard English?

Reviewer #2: Yes

6. Review Comments to the Author

Reviewer #2: Most reviewer comments have been considered, and the paper has been reworked and rewritten. So, the paper is amended, but some issues still need to be solved.

Table 3 is clear, but what does the reference figure highlight? If 37 articles are used to get the quotations, what are the references?—171?

Another area for improvement is how I, as a reader, will judge or understand this review's quality.

There are no or limited references presenting the different articles included in the result, and there are no figures about how many articles built the domains- Table 3 gives some numbers- but there could be repeated articles. Since we do not know how many of the included 51 articles/reports used in this review are used/presented in every domain.

In the secondary objective/result, you present some more references—Fifteen articles (39,47–49,52,58,63,64,66,67,72,78–80,83) specifically explored the role of….Nevertheless, mostly, there is no information, and the text presented does not show any evidence that it is coming from the articles when just presenting a few quotations- for example, 4 quotations from 51 articles.

The text is supposed to be the analysis from the articles/quotations—the text extraction.

Forty-three studies explored coping strategies and positive action; please add the references.

The discussion, well, there is a result presentation- those references presenting the result should be in the result section, and you are supposed to discuss your findings from your review against articles/references from your background or new ones.

There are 23 references from your review result in the discussion- so were only 23 out of 51 articles used to build the result? 3 references from the introduction are found in the discussion and 5 "new" ones.

Still, there needs to be some clarification: in the abstract, you state that 51 articles/reports were included in the review- references 39-90 are those articles. However, are 91-93 included in the result as well?

In the implications, there are references from the introduction, the result, and a new one. So, what are your implications from the result?

Limitations of the included articles—the quality of those articles should be judged during the appraisal phase.

The conclusion is that this is out of the review. There are several aspects I cannot recall from the review results.

Still, some years are missing in reference 5—it should be 2021, and reference 102 should be 2022.

As I said before, This manuscript would contribute to our knowledge base if the data were analysed and handled according to the references.

7. PLOS authors have the option to publish the peer review history of their article (what does this mean?). If published, this will include your full peer review and any attached files.

Reviewer #2: No

---

## [Author Response · Author response to Decision Letter 1]

16 Apr 2024

Reviewer’s comments to the authors:

Journal Requirements:

We reviewed the references list to ensure that any retracted references are removed and replaced. 

Reviewer 2 comments:

4. Have the authors made all data underlying the findings in their manuscript fully available?

Reviewer #2: No

All data we are able to provide are uploaded in the form of supplementary data tables and selected quotes. This is a review, so it would not be possible or appropriate to include the entire contents of the included papers before asking permission to authors with great delay in the submission of this paper to this journal. All this information is available from the selected and referenced original papers. 

To support transparency, we have added the updated protocol to the supplementary materials to ensure the most up-to-date version is available to all readers. 

Reviewer #2: Most reviewer comments have been considered, and the paper has been reworked and rewritten. So, the paper is amended, but some issues still need to be solved.

Table 3 is clear, but what does the reference figure highlight? If 37 articles are used to get the quotations, what are the references?—171?

The word ‘’reference’’ in table 3 has been given further explanation in the text.

Another area for improvement is how I, as a reader, will judge or understand this review's quality.

There are no or limited references presenting the different articles included in the result, and there are no figures about how many articles built the domains- Table 3 gives some numbers- but there could be repeated articles. Since we do not know how many of the included 51 articles/reports used in this review are used/presented in every domain.

The articles numbers in table 3 do show how many of the 51 articles have made reference to each domain. In some areas, we had not cited all papers (e.g. for categories and coping strategies) because we reported a summary of findings, but we now have ratified this, more references have been added. 

In the secondary objective/result, you present some more references—Fifteen articles (39,47–49,52,58,63,64,66,67,72,78–80,83) specifically explored the role of….Nevertheless, mostly, there is no information, and the text presented does not show any evidence that it is coming from the articles when just presenting a few quotations- for example, 4 quotations from 51 articles.

Table 4 shows an example of quotations for each domain. We have changed the Table description to make this clear and added more citations to individual studies in the description of themes and categories. We are not sure that presenting citations from all 51 articles will make the narrative easier for readers, and it is a secondary objective. We summarized the information deriving from quotes to make the content accessible and in line with other qualitative reviews. 

The text is supposed to be the analysis from the articles/quotations—the text extraction.

It is; we focused on a description of the themes, since using quotes from 51 articles may not be efficient or easy to read or follow. In the previous draft we inserted the quotes within the text, and it was difficult to follow and understand the main points we were trying to portray. We looked at published literature and decided to insert a table with selected quotes as the best approach we have seen so far.

Forty-three studies explored coping strategies and positive action; please add the references.

These are now listed and the number corrected.

The discussion, well, there is a result presentation- those references presenting the result should be in the result section, and you are supposed to discuss your findings from your review against articles/references from your background or new ones.

We referred to the results in the discussion as thought to be good practice, to link the narrative from one section to another. We introduced some new reference to make clear to the reader that we didn’t infer information pertaining the construct of moral distress from the papers we reviewed, but we focused our analysis of quotations (as explained in methods) to describe triggers and factors. In the introduction we ‘’introduced’’ the topic from a psychological/philosophical point of view drawing from experts’ opinions, then we went beyond that, understanding the triggers and factors producing moral distress in workers to offer practical insights. 

There are 23 references from your review result in the discussion- so were only 23 out of 51 articles used to build the result? 3 references from the introduction are found in the discussion and 5 "new" ones.

We understand the listing of 23 references might be confusing, we erased it and made clear that all 51 articles were used in the review, and all are cited. We have clarified this in a few places to make more accessible how this was achieved and where each was used. New references were used when necessary to suggest the reader of potential evidence-based consequences on quality of care and potential solutions that have been found to be effective.

Still, there needs to be some clarification: in the abstract, you state that 51 articles/reports were included in the review- references 39-90 are those articles. However, are 91-93 included in the result as well?

We have checked these references specifically and have ensured that they are correctly cited where relevant. 91 refers to a book used for definitions, this is only cited where this is relevant. 92 is included and cited appropriately now. 93 was mistakenly left in but has now been removed. 

In the implications, there are references from the introduction, the result, and a new one. So, what are your implications from the result?

The discussion starts with reference to the construct of moral distress and injury, linking the narrative to the introduction where the concept of moral distress is discussed and adding the definition of moral reasoning (new reference) to provide deeper understanding of different aspects of the problem. We then move to the description of the triggers and factors of distress as emerging from the results, therefore a comparison between the introduction references and the results findings is limited for obvious reasons. The implications from the results are clearly stated in the specific sections. 

Limitations of the included articles—the quality of those articles should be judged during the appraisal phase.

This was done and is detailed in the method, results and in table 1. In line with recommendations for structuring of systematic review articles (Page, M.J., McKenzie, J.E., Bossuyt, P.M. et al. The PRISMA 2020 statement: an updated guideline for reporting systematic reviews. Syst Rev 10, 89 (2021). https://doi.org/10.1186/s13643-021-01626-4). An additional summary of the key issues is highlighted here to help readers know where there is need for improvement in this body of research. 

The conclusion is that this is out of the review. There are several aspects I cannot recall from the review results.

Similarly to other points made by this reviewer, the authors struggled to understand the meaning of this comment (is referring to the conclusion of the limitations named above – if so, such conclusion was not made by us) We have inferred that the reviewer thinks some of the conclusions do not accurately reflect, or go beyond what was presented in the results We have addressed this, to better explain and highlight the link between our results and the conclusion. 

Still, some years are missing in reference 5—it should be 2021, and reference 102 should be 2022.

We have corrected this.

As I said before, this manuscript would contribute to our knowledge base if the data were analysed and handled according to the references.

A number of appropriate methodology papers have been cited and the process of analysis was followed when compiling this qualitative review and, in our view, thoroughly explained. Having looked at other published PloS qualitative reviews (e.g. Iyahen EO, Omoruyi OO, Rowa-Dewar N, Dobbie F (2023). Exploring the barriers and facilitators to the uptake of smoking cessation services for people in treatment or recovery from problematic drug or alcohol use: A qualitative systematic review. PLoS ONE 18(7): e0288409.), we are confident that our paper provides a similar degree of detail and insight to the processes followed.

---

## [Editor Report · Decision Letter 2]

18 Apr 2024

Triggers and factors associated with moral distress and moral injury in health and social care workers: a systematic review of qualitative studies

PONE-D-23-19884R2

Dear Dr. Beadle,

We’re pleased to inform you that your manuscript has been judged scientifically suitable for publication and will be formally accepted for publication once it meets all outstanding technical requirements.

Kind regards,

Nancy Clark, PhD

Academic Editor

PLOS ONE
---

## [Editor Report · Acceptance letter]

13 May 2024

PONE-D-23-19884R2 

PLOS ONE

Dear Dr. Beadle, 

I'm pleased to inform you that your manuscript has been deemed suitable for publication in PLOS ONE. Congratulations! Your manuscript is now being handed over to our production team.

Kind regards, 

on behalf of

Dr. Nancy Clark 

Academic Editor

PLOS ONE